# LOCALITY-ATTENDING VISION TRANSFORMER

**Sina Hajimiri**✉**,    Farzad Beizaee,    Fereshteh Shakeri,**
**Christian Desrosiers,    Ismail Ben Ayed,    Jose Dolz**
ÉTS Montreal, LIVIA, ILLS
✉ seyed-mohammadsina.hajimiri.1@etsmtl.net

## ABSTRACT

Vision transformers have demonstrated remarkable success in classification by leveraging global self-attention to capture long-range dependencies. However, this same mechanism can obscure fine-grained spatial details crucial for tasks such as segmentation. In this work, we seek to enhance segmentation performance of vision transformers after standard image-level classification training. More specifically, we present a simple yet effective add-on that improves performance on segmentation tasks while retaining vision transformers' image-level recognition capabilities. In our approach, we modulate the self-attention with a learnable Gaussian kernel that biases the attention toward neighboring patches. We further refine the patch representations to learn better embeddings at patch positions. These modifications encourage tokens to focus on local surroundings and ensure meaningful representations at spatial positions, while still preserving the model's ability to incorporate global information. Experiments demonstrate the effectiveness of our modifications, evidenced by substantial segmentation gains on three benchmarks (*e.g.*, over $6\%$ and $4\%$ on ADE20K for ViT Tiny and Base), without changing the training regime or sacrificing classification performance. The code is available at https://github.com/sinahmr/LocAtViT/.

## 1  INTRODUCTION

Vision transformers (ViT, Dosovitskiy et al., 2021) have emerged as powerful visual backbones by modeling images as sequences of patch tokens, processed with self-attention. Unlike convolutional neural networks (CNN, LeCun et al., 2015), which aggregate local information in a restricted receptive field, ViTs can capture long-range dependencies at any layer. This global attention mechanism has proven highly effective for image classification, enabling ViT models to surpass CNN performance when sufficient data is available (Touvron et al., 2021a). A key factor behind this success is the ability to integrate global context that leads to more uniform and holistic representations across layers, which enhances the recognition of high-level image semantics (Raghu et al., 2021).

The same global focus that makes ViTs excel in classification, however, poses challenges for dense prediction tasks such as semantic segmentation. These tasks require precise localization and fine-grained spatial detail, properties that convolutional inductive biases naturally encourage but vanilla ViTs lack (Hassani et al., 2023). As a result, the design of spatial attention and feature hierarchy has been found critical for adapting transformers to dense tasks (Wang et al., 2021; Liu et al., 2021). Still, a tension remains between capturing global context and preserving local detail. Global attention can dilute local cues, whereas purely local schemes may miss long-range dependencies needed for holistic understanding. Besides, the classification objective used by models often neglects the necessities of dense prediction, motivating a need for a "segmentation-in-mind" pretraining. Empirically, we show in Appendix F that, in a ViT trained for classification, patch tokens progressively lose distinct local structure and become increasingly aligned with the [CLS] token.

More recently, foundation models trained at large-scale (Radford et al., 2021; Oquab et al., 2024), which learn versatile visual representations, have seen broad adoption in a breadth of visual tasks. Despite the availability of more intricate designs, these models still mostly adopt vanilla ViT due to its simplicity and ease of integration. This widespread reliance underscores the practical value of enhancing ViT's capabilities rather than pursuing more complex new designs. A prominent example is CLIP (Radford et al., 2021), which couples a ViT-based image encoder with a text encoder

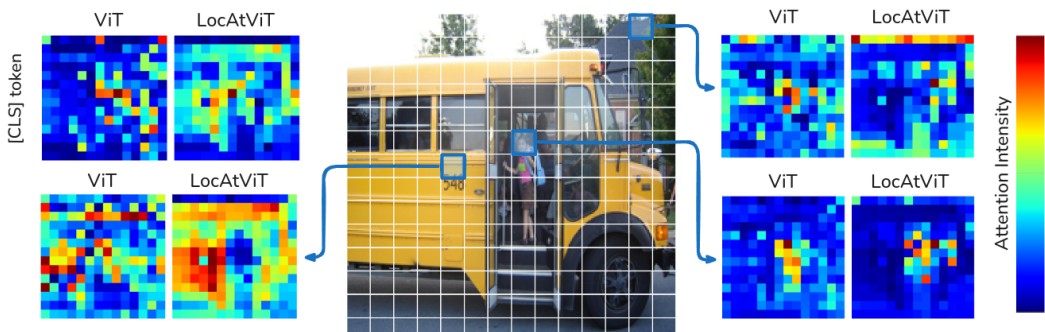

Figure 1: **Qualitative evaluation on the attention maps.** The final attention maps (before the classification head) of ViT and LocAtViT for the `[CLS]` token and three patches are illustrated for an image with label *school bus*.

to align representations, enabling zero-shot classification and open-vocabulary recognition. Such representations can be repurposed for dense prediction, for instance, by comparing local features to text prompts, but this adaptation is non-trivial. Furthermore, recent studies try to harness CLIP's knowledge for segmentation without any task-specific training (Zhou et al., 2022; Wang et al., 2024; Hajimiri et al., 2025). However, as CLIP and similar models are not trained for quality local representations, their features often lack the spatial granularity needed for precise dense prediction.

**Contributions.** In this paper, we propose a modular *Locality-Attending* (LocAt) add-on, which incorporates two ideas: *(i)* We modulate the attention logits with a learnable Gaussian kernel centered on each token's location, ensuring that patches closer to the token receive higher attention. This acts as an explicit inductive bias encouraging each token to attend to its local neighborhood while still allowing global interactions. We denote the resulting self-attention module as the *Gaussian-Augmented* (GAug) attention (Section 4.1). *(ii)* We enhance patch representations for segmentation by introducing minor changes prior to the classification head, preserving the meaningfulness of spatial tokens that are most important for dense prediction. We term this procedure *Patch Representation Refinement* (PRR) that addresses the gradient flow issue in ViTs for segmentation, which is overlooked in the literature (see Section 4.2). LocAt refers to the combination of GAug and PRR, and Figure 2 demonstrates that it improves different baselines, yielding significant segmentation performance gains (arrows pointing

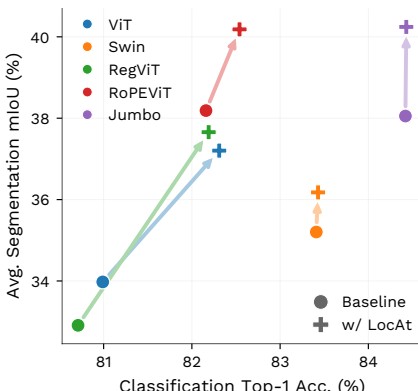

Figure 2: **LocAt considerably enhances different baselines** in segmentation, while preserving or even improving classification.

upward), while preserving or improving classification accuracy (no arrow pointing to the left). The proposed add-on also enhances the quality of attention maps, as illustrated in Figure 1. LocAt is a lightweight and objective-agnostic add-on, also compatible with self-supervised pretraining. Importantly, the minimal architectural changes required to integrate LocAt make it readily applicable to any ViT with marginal changes, facilitating its usage in foundation models. Our perspective is that ViT pretraining should be designed with downstream dense prediction in mind, while remaining faithful to the vanilla ViT architecture and training regime.

## 2 RELATED WORK

**Hierarchical ViT backbones for dense prediction.** While the original ViT targets image classification and produces low-resolution features with weak locality priors (Dosovitskiy et al., 2021), dense prediction has motivated backbones that retain or recover spatial detail across stages. Some

works use pyramid and token-merging designs to introduce multi-scale features and lightweight decoders for segmentation (Wang et al., 2021; Xie et al., 2021), while others build parallel branches for local and global processing (Chu et al., 2021). These works show that topology substantially helps dense tasks. However, they typically require non-trivial architectural changes (new stages or merging blocks) and may rely on local window attention that limits full-image interaction.

**Convolution-based hybrids.** Another line injects convolutional priors either inside attention or in the feed-forward network to encourage local bias while keeping global modeling. Works use convolutional projections (Wu et al., 2021a), add gated positional self-attention to softly bias toward convolutional behavior (d'Ascoli et al., 2021), couple local convolutional features with global representations (Peng et al., 2021), or add convolutions in the feed-forward network (Li et al., 2021). These hybrid models add extra modules that require tuning, and they can reduce plug-and-play compatibility with off-the-shelf ViTs, as they often introduce branches or replace core components. Besides, convolution offers a spatially-shared kernel which is independent of patch information.

**Locality mechanisms inside attention.** Orthogonal to backbone design, many papers modify the attention pattern itself to introduce locality. Many of the works use fixed or structured windows (Liu et al., 2021; Dong et al., 2022; Yang et al., 2021). Other ideas include utilizing sliding or dilated neighborhoods to expand receptive fields efficiently (Hassani et al., 2023; Hassani & Shi, 2022), sampling content-relevant keys (Xia et al., 2023), selecting regions using dynamic sparse routing (Zhu et al., 2023), or using explicit global-local mixers to balance context with locality (Ding et al., 2022; Tu et al., 2022; Chen et al., 2022; Hatamizadeh et al., 2023). Most of these approaches restrict or mask interactions (using windows or patterns) or add mixing subsystems that complicate design, impeding their widespread adoption.

**Positional encodings that strengthen locality.** Beyond absolute embeddings, relative positional encoding (RPE), and rotary positional encodings (RoPE) improve spatial awareness in ViTs (Shaw et al., 2018; Liu et al., 2021; Wu et al., 2021b; Su et al., 2024; Heo et al., 2024). These approaches are orthogonal to attention locality, and we briefly mentioned them to emphasize that they encode locality as well. Our work complements rather than replaces them, as we show in the experiments.

**Improving token representation.** Recent work on register tokens augments ViTs with dedicated auxiliary tokens that absorb non-informative computation and yield smoother feature maps helpful for dense prediction (Darcet et al., 2024). Unlike this approach, we do not require auxiliary tokens, and we also address the issue of gradient flow to spatial patch outputs, overlooked in the prior work. Some works introduce class-attention layers that specialize the last blocks to refining only the class token, while keeping patch tokens fixed in those layers, leading to suboptimal dense prediction performance (Touvron et al., 2021b). Finally, pooling heads such as global average pooling (GAP) and multihead attention pooling (Zhai et al., 2022) aim to produce a stronger pooled representation for classification by aggregating patch tokens, while our work is explicitly designed for segmentation-in-mind training with an emphasis on improving the spatial token representations themselves rather than only the pooled vector.

**Foundation models for dense prediction.** Large pre-trained foundation models, such as CLIP (Radford et al., 2021), demonstrate impressive zero-shot generalization on image-level recognition by leveraging ViT backbones. The preference for the standard ViT backbone can be attributed to its strong global attention, predictable scaling behavior with data and model size, and a uniform architecture that avoids the need for complex stage-wise tuning as the model grows (Zhai et al., 2022; Alabdulmohsin et al., 2023). However, despite excelling on image-level benchmarks, such models remain less effective for dense prediction because their representations are predominantly global and task-agnostic (Shao et al., 2024). As a result, additional adaptation or decoding layers are usually required to repurpose them for segmentation or detection (Li et al., 2022; Xu et al., 2023; Luo et al., 2023). While these adaptations yield improvements, they do not fully address the core issue: foundation-model ViTs—trained with classification objectives—tend to emphasize global semantics over local detail (Liang et al., 2023).

A ViT backbone that natively preserves both local detail and global context could enable foundation models to excel at dense prediction without extra adaptation layers or specialized fine-tuning. In this work, we take a step in that direction by refining the ViT backbone itself. Our approach aims to

potentially bridge the gap between the powerful image-level understanding and the requirements of pixel-level prediction tasks.

## 3 PRELIMINARIES

Each ViT layer $l$ takes a sequence of tokens $\mathbf{x}^{(l-1)} \in \mathbb{R}^{(1+hw)\times C}$ as input, containing a `[CLS]` token and $hw$ spatial patch tokens. Each token is a $C$-dimensional vector, and $h$ and $w$ denote the number of patches in each column and row. $\mathbf{x}^{(0)}$ is the partitioned and flattened input after adding the positional embeddings. At each layer $l$, the following operations are applied, where LN, attn, and MLP denote layer normalization, self-attention, and feed-forward network, respectively:

$$\mathbf{x}' = \mathbf{x}^{(l-1)} + \text{attn}\Big(\text{LN}(\mathbf{x}^{(l-1)})\Big), \tag{1}$$

$$\mathbf{x}^{(l)} = \mathbf{x}' + \text{MLP}\Big(\text{LN}(\mathbf{x}')\Big). \tag{2}$$

Each self-attention module (attn) consists of two sets of weight matrices: $\mathbf{W}^q, \mathbf{W}^k, \mathbf{W}^v \in \mathbb{R}^{C\times d}$ to compute $d$-dimensional query, key, and value matrices (*i.e.*, $\mathbf{q}, \mathbf{k}, \mathbf{v} \in \mathbb{R}^{(1+hw)\times d}$) based on the input, and $\mathbf{W}^o \in \mathbb{R}^{d\times C}$ for the final projection. After obtaining $\mathbf{q}$, $\mathbf{k}$, and $\mathbf{v}$, we calculate:

$$\mathbf{Z} = \text{softmax}\Big(\mathbf{q}\mathbf{k}^\top/\sqrt{d}\Big)\mathbf{v}. \tag{3}$$

Matrix $\mathbf{Z} \in \mathbb{R}^{(1+hw)\times d}$ is then transformed by $\mathbf{W}^o$ to form the output of the layer. The *attention logits* of a patch $p$ are represented by the $p^{\text{th}}$ row of $\mathbf{q}\mathbf{k}^\top/\sqrt{d}$. Note that for simplicity, we present the formulation of a single-head self-attention.

## 4 METHOD

We now present **LocAtViT**, which enhances ViT with two modular components, GAug attention (Section 4.1) and PRR (Section 4.2), and is trained with the same classification objective as ViT.

### 4.1 GAUSSIAN-AUGMENTED ATTENTION

We introduce explicit locality into vision transformers by adding a patch-specific Gaussian kernel to the attention logits for all spatial tokens. We first present the modified self-attention, then describe how the kernel is computed, and finally define the resulting attention addition.

**Modified self-attention.** At each self-attention layer, we add a *supplement* matrix $\mathbf{S}$ to the attention logits, encouraging each patch to attend more strongly to its local neighborhood. With this addition, the self-attention formulation of Eq. (3) is modified as follows, which is also depicted in Figure 3a:

$$\mathbf{Z} = \text{softmax}\left(\frac{\mathbf{q}\mathbf{k}^\top}{\sqrt{d}} + \mathbf{S}\right)\mathbf{v}. \tag{4}$$

We construct $\mathbf{S}$ so that a patch $p$ attends more to its immediate surroundings, with the added bias decaying smoothly with distance from $p$. Since our locality prior is defined on the spatial grid, it does not apply to the `[CLS]` token (which has no spatial coordinates). Concretely, we first compute all the following quantities for the $hw$ spatial tokens, and then embed them into a $(1+hw)\times(1+hw)$ matrix by zero-padding the row and column corresponding to `[CLS]`.

A natural choice for such a distance-based locality prior is an unnormalized Gaussian centered at $p$ (more information is available in Appendix E). A Gaussian kernel provides a smooth, monotone decay of influence with distance, controlled by a variance parameter $\sigma^2$ (in the isotropic case). This gives an interpretable handle on the effective receptive field: small $\sigma$ yields a sharp, highly local focus, whereas large $\sigma$ approaches a nearly uniform weighting over patches.

We parameterize the variance of the Gaussian kernel for each patch by a 2D vector, stored in the $p^{\text{th}}$ row of $\mathbf{\Sigma} \in \mathbb{R}_+^{hw\times 2}$, which controls the attention span along both axes. Because patches may

require different receptive fields, we predict these variances from the *spatial* query matrix, *i.e.*, $\mathbf{q}_{\mathrm{sp}} \in \mathbb{R}^{hw \times d}$, the sub-matrix of $\mathbf{q}$ obtained by removing the `[CLS]` row. Using a learnable weight matrix $\mathbf{W}^{\sigma} \in \mathbb{R}^{d \times 2}$ (with $f$ a scaled sigmoid ensuring positive, bounded values), we compute:

$$\mathbf{\Sigma} = f(\mathbf{q}_{\mathrm{sp}}\mathbf{W}^{\sigma}). \tag{5}$$

**Gaussian kernel.**   For a patch grid of size $h \times w$, we denote the set of coordinate vectors as:

$$\mathbf{P} = \begin{bmatrix} i & j \end{bmatrix}_{i \in \{1,2,...,h\},\ j \in \{1,2,...,w\}}, \tag{6}$$

in an $hw \times 2$ matrix. The $hw \times hw \times 2$ pairwise squared difference $\mathbf{D}$ is computed as:

$$\mathbf{D}_{ptm} = \left( \mathbf{P}_{pm} - \mathbf{P}_{tm} \right)^2, \quad \text{for } m \in \{1, 2\}, \tag{7}$$

where $p$ and $t$ denote indices of the source and target patches, and $m$ indexes the coordinate dimensions. Given $\mathbf{\Sigma}$, we compute the Gaussian kernel over spatial tokens, $\mathbf{G} \in \mathbb{R}_+^{hw \times hw}$, as:

$$\mathbf{G}_{pt} = \exp\left(-\frac{1}{2}\sum_{m=1}^{2} \frac{\mathbf{D}_{ptm}}{\mathbf{\Sigma}_{pm}}\right), \tag{8}$$

which determines the addition to attention logits from patch $p$ to $t$.

**Supplement matrix.**   From Eq. (8), each entry of $\mathbf{G}$ lies in $[0, 1]$, therefore, directly adding $\mathbf{G}$ to the attention logits would create a scale mismatch. To mitigate this, we use a learnable weight matrix $\mathbf{W}^{\alpha} \in \mathbb{R}^{d \times 1}$ that predicts a per-query scaling from the spatial query matrix. Entries in $\boldsymbol{\alpha}$ scale rows of the Gaussian kernel (softplus ensures positive coefficients), which yield $\mathbf{S}$ after zero-padding the `[CLS]` row and column. Concretely:

$$\boldsymbol{\alpha} = \mathrm{softplus}(\mathbf{q}_{\mathrm{sp}}\mathbf{W}^{\alpha}) \in \mathbb{R}_+^{hw}, \tag{9}$$

$$\mathbf{S} = \begin{bmatrix} 0 & \mathbf{0}^{\top} \\ \mathbf{0} & \mathrm{diag}(\boldsymbol{\alpha})\,\mathbf{G} \end{bmatrix} \in \mathbb{R}^{(1+hw) \times (1+hw)}. \tag{10}$$

Intuitively, $\boldsymbol{\alpha}$ acts as a per-query, row-wise balancing factor between the original attention logits and the Gaussian locality prior. For tokens where the network predicts small values of $\boldsymbol{\alpha}$, the contribution of $\mathbf{S}$ is negligible and the behavior approaches standard global self-attention (weak locality), whereas larger values of $\boldsymbol{\alpha}$ yield a stronger local bias. This makes our approach a soft, data-dependent locality mechanism rather than a hard constraint. We empirically analyze the effect of this scaling, as well as parameter-free alternatives, in Appendix D.3 and D.4.

We refer to our modified self-attention as *Gaussian-Augmented* (GAug) attention. Figure 3b illustrates the generation process of the supplement matrix.

## 4.2   Patch Representation Refinement

**Problem statement.**   In a classification task using ViT, only the `[CLS]` token's output of the model is used for computing the loss. While effective for classification, this approach has fundamental limitations for dense prediction from a gradient flow perspective. More concretely, the patch positions' outputs receive no *direct* supervision, *i.e.*, it is not important to the model what ViT's final outputs are at those positions. However, these output representations are crucial for further dense prediction. This is problematic because the fine-grained spatial information carried by individual patch tokens is not effectively learned at the final layer.

Some subsequent methods, such as Swin (Liu et al., 2021), remove the `[CLS]` token and use global average pooling (GAP) before the classification head. However, this forces an undesirable behavior from a dense prediction standpoint, *i.e.*, a *uniform gradient flow* across all positions. For example, in an image of a bird with other objects in the background, GAP compels the model to match all patch representations—including background regions—with the classifier's prototype of bird. The uniform gradient flow means that all patch tokens receive equal importance, regardless of their relevance, potentially leading to representations particularly suboptimal for tasks like segmentation. Moreover, GAP has been shown to reduce localization in higher layers (Raghu et al., 2021).

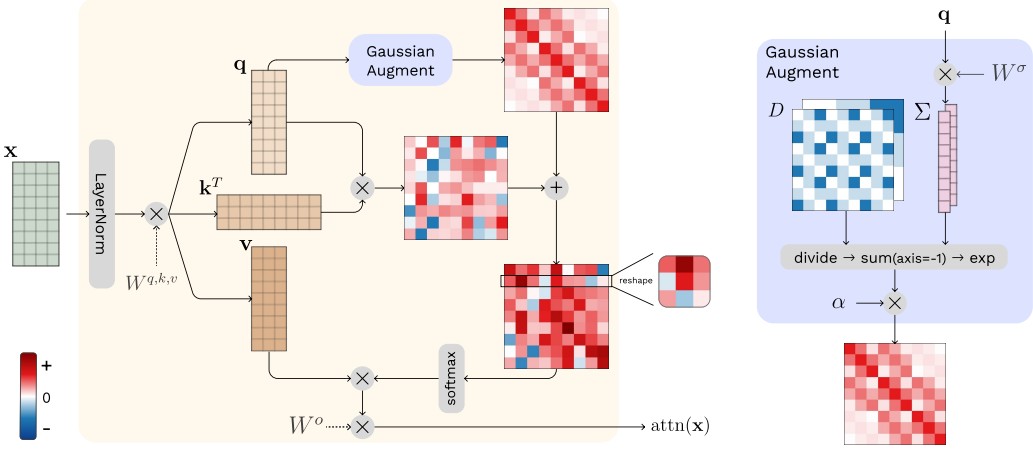

(a) Modified self-attention.

(b) Supplement matrix $\mathbf{S}$.

Figure 3: **Illustration of the Gaussian-Augmented attention** for a $3 \times 3$ grid. *(a)* The Gaussian addition is obtained based on the query and is added to the attention logits. The $p^{\text{th}}$ row in the attention logits matrix presents the attention of patch $p$ to all patch tokens. The reshaped matrix illustrates that with GAug, both local and global attentions are integrated. *(b)* The supplement matrix $\mathbf{S}$ encourages attending to the locality and is computed using the pairwise squared difference tensor $\mathbf{D}$ from Eq. (7). For simplicity, the [CLS] token is not shown in this visualization, and Gaussian variances and scaling coefficients are set to a constant value for all patches.

**Proposed solution.** To encourage meaningful patch representations at the final layer's output, $\mathbf{x} \in \mathbb{R}^{(1+hw) \times C}$, we apply a parameter-free attention before the classification head. We reshape $\mathbf{x}$ into $H$ heads, $\mathbf{x} \to \{\mathbf{x}_i\}_{i=1}^H$, where $\mathbf{x}_i \in \mathbb{R}^{(1+hw) \times d}$ and compute:

$$\mathbf{x}_i^+ = \mathrm{softmax}\left(\frac{\mathbf{x}_i \mathbf{x}_i^\top}{\sqrt{d}}\right) \mathbf{x}_i \,, \tag{11}$$

then reshape back to $\mathbf{x}^+ \in \mathbb{R}^{(1+hw) \times C}$. This can be viewed as a *parameter-free* multi-head self-attention. This operation, which introduces no new parameters, aggregates information from all patch positions in a non-uniform manner, thereby preserving their unique contributions and ensuring diverse gradient flow across patch locations. The resulting representation at the [CLS] token, $\mathbf{x}_0^+$, is then passed to the classification head. We refer to this strategy as *Patch Representation Refinement* (PRR), which can be seen as an alternative to GAP, suitable for segmentation-in-mind pretraining.

Our components share the common objective of making ViT's representations more suitable for dense prediction, and they act at different stages. GAug operates inside the backbone, modifying self-attention to bias information exchange toward local neighborhoods so that patch tokens can better encode fine spatial details. PRR, in contrast, acts right before the classification head and changes how tokens are aggregated to explicitly route supervision and gradients to patch outputs. In practice, each module can be attached independently to a ViT backbone (see ablations in Section 5.4). However, they are coupled through the gradient path: with standard [CLS] classification, if PRR is not present, last block's GAug has little effect because its parameters receive no gradient from the loss. PRR routes gradients to those GAug parameters so they can be effectively learned.

## 5 EXPERIMENTS

### 5.1 EXPERIMENTAL SETUP

**Datasets.** For the main experiments, where we assess both classification and segmentation performance, we first train models on ImageNet-1K (Deng et al., 2009; Russakovsky et al., 2015), which contains 1.28M training images from 1,000 classes. Then, we further utilize these models for training on segmentation datasets: ADE20K (ADE, Zhou et al., 2019), PASCAL Context (P-Context, Mottaghi et al., 2014), and COCO Stuff (C-Stuff, Caesar et al., 2018; Lin et al.,

2014), which contain 150, 59, and 171 semantic categories, respectively. ADE20K and COCO Stuff images are resized to $512 \times 512$ and PASCAL Context images to $480 \times 480$. Furthermore, we also assess classification performance on smaller scale datasets: CIFAR-100 (Krizhevsky & Hinton, 2009) and mini-ImageNet (Vinyals et al., 2016), a subset of ImageNet-1K, consisting of 100 classes with 500 training and 100 validation examples each. In all classification experiments, images are resized to $224 \times 224$.

**Implementation details.** Our method is implemented using the PyTorch Image Models (`timm`) (Wightman, 2019) library. We train models on ImageNet-1K for 300 epochs, with initial learning rate (LR) 0.001 and 20 epochs of linear warm-up. CIFAR-100 and mini-ImageNet models are trained for 600 epochs, with LR 0.0005 and 120 epochs of linear warm-up. Global batch size is set to 1024, and we use AdamW (Kingma & Ba, 2015; Loshchilov & Hutter, 2019) optimizer with a weight decay of 0.05. Similar to Ding et al. (2022), a simple triangular learning rate scheduler (Smith & Topin, 2018) is applied, and the stochastic depth drop rates (Huang et al., 2016) for the Tiny, Small, and Base backbones are set to 0.1, 0.2, and 0.4, respectively. We follow Liu et al. (2021) for data augmentation and use RandAugment (Cubuk et al., 2020), Mixup (Zhang et al., 2018), Cutmix (Yun et al., 2019), and random erasing (Zhong et al., 2020). The sigmoid function $f$ in Eq. (5) is scaled to have a maximum of $\max(h, w)$, and shifted to satisfy $f(0) = 1$.

For semantic segmentation, we utilize the MMSegmentation toolbox (OpenMMLab, 2020) and employ a simple 1-layer MLP on top of the frozen classification-trained models. This configuration ensures that segmentation performance mainly reflects the discriminative power of the classification-trained backbones in dense prediction. This setup aligns with our goal of isolating and assessing patch representation quality under a low-tuning regime. Training on segmentation datasets is performed over 20K iterations with a batch size of 32. When processing images at resolutions different from the pretraining resolution, we scale GAug's variance proportionally.

## 5.2 MAIN RESULTS

**Segmentation performance.** The LocAt add-on can be applied to several ViT-based models, and Table 1 evaluates its effect, in terms of classification performance on ImageNet-1K, as well as segmentation performance on three benchmarks, when applied to five models: ViT (Dosovitskiy et al., 2021), Swin Transformer (Liu et al., 2021), ViTs with registers (denoted as RegViT, we use 4 registers, Darcet et al., 2024), Rotary Position Embedding for ViTs (denoted as RoPEViT, Heo et al., 2024), and Jumbo (Fuller et al., 2026). Comparing each baseline with its enhanced counterpart (gray row below), indicates that LocAt's addition is useful in improving the segmentation performance of all. For instance, LocAtViT Tiny achieves a substantial improvement of **+6.17%**, **+4.86%**, and **+5.86%**, over ViT on ADE20K, PASCAL Context, and COCO Stuff, respectively. Importantly, LocAt-enhanced models' superior segmentation performance is achieved without compromising classification performance; in fact, they deliver comparable or even improved accuracy across different models (*e.g.*, LocAtViT outperforms ViT by **+1.55%** in the Tiny backbone).

LocAt improves baselines that are architecturally close to ViT significantly, *e.g.*, RoPEViT, and interestingly, it brings improvements over Swin as well. We believe this is not trivial as the add-on was designed for ViT's architecture, in which there exists a `[CLS]` token and the attention width is not limited, while in Swin the windowed attention mechanism severely affects the extent to which LocAt can play a role. Furthermore, our add-on incurs a negligible increase in computational cost in terms of number of FLOPs over the corresponding counterparts (measured at $224 \times 224$ using Sovrasov, 2018). Additional experiments are presented in Appendix B.

**Classification performance.** In addition to the ImageNet-1K classification results in Table 1, Table 2 investigates LocAt's classification effectiveness on small-scale datasets: mini-ImageNet (Vinyals et al., 2016) and CIFAR-100 (Krizhevsky & Hinton, 2009). Although designed to enhance segmentation, these results demonstrate LocAt's classification effectiveness even when trained on small-scale datasets. LocAt improves ViT's performance by 3-6% on mini-ImageNet and 4-7% on CIFAR-100, while only introducing 2,340 new parameters (0.003% increase for Base). We do not report segmentation results for models trained on these datasets since due to their scale and number of classes, representations are not expected to generalize well to segmentation benchmarks.

Table 1: **Segmentation performance** of models and their counterparts with our LocAt extension (in gray), along with their **classification performance** on ImageNet-1K, which the models are initially trained on. Results demonstrate that *(i)* LocAt substantially boosts segmentation performance (*our primary focus*), while preserving or even improving the classification performance, and *(ii)* this effect holds for a variety of methods, for different backbone sizes. Furthermore, *(iii)* the segmentation gains appear not only in weaker baselines, but also in strong, high-performing models, where classification improvements are harder to achieve.

| | Method | Segmentation mIoU (%) | | | Top-1 (%) | #Params | FLOPs |
|---|---|---|---|---|---|---|---|
| | | ADE | P-Context | C-Stuff | ImageNet | (M) | (G) |
| **Tiny** | ViT | 17.30 | 33.71 | 20.29 | 72.39 | 6 | 1.26 |
| | + LocAt | $23.47_{+6.17}$ | $38.57_{+4.86}$ | $26.15_{+5.86}$ | $73.94_{+1.55}$ | 6 | 1.27 |
| | Swin | 25.58 | 36.78 | 28.34 | 81.18 | 28 | 4.50 |
| | + LocAt | $26.52_{+0.94}$ | $37.65_{+0.87}$ | $29.09_{+0.75}$ | $81.43_{+0.25}$ | 28 | 4.51 |
| | RegViT | 15.98 | 33.45 | 19.58 | 72.90 | 6 | 1.29 |
| | + LocAt | $24.39_{+8.41}$ | $39.90_{+6.45}$ | $27.38_{+7.80}$ | $74.08_{+1.18}$ | 6 | 1.30 |
| | RoPEViT | 19.17 | 38.16 | 22.75 | 73.60 | 6 | 1.26 |
| | + LocAt | $24.48_{+5.31}$ | $40.79_{+2.63}$ | $27.98_{+5.23}$ | $74.34_{+0.74}$ | 6 | 1.27 |
| | Jumbo | 20.33 | 36.36 | 22.13 | 78.71 | 17 | 1.40 |
| | + LocAt | $21.62_{+1.29}$ | $37.22_{+0.86}$ | $23.87_{+1.74}$ | $78.78_{+0.07}$ | 17 | 1.42 |
| **Base** | ViT | 28.40 | 43.10 | 30.43 | 80.99 | 86 | 17.58 |
| | + LocAt | $32.64_{+4.24}$ | $45.35_{+2.25}$ | $33.62_{+3.19}$ | $82.31_{+1.32}$ | 86 | 17.64 |
| | Swin | 31.90 | 40.11 | 33.60 | 83.41 | 88 | 15.46 |
| | + LocAt | $32.89_{+0.99}$ | $41.44_{+1.33}$ | $34.20_{+0.60}$ | $83.43_{+0.02}$ | 88 | 15.47 |
| | RegViT | 27.93 | 41.81 | 28.99 | 80.71 | 86 | 17.95 |
| | + LocAt | $32.71_{+4.78}$ | $46.14_{+4.33}$ | $34.12_{+5.13}$ | $82.19_{+1.18}$ | 86 | 18.02 |
| | RoPEViT | 31.38 | 48.83 | 34.35 | 82.16 | 86 | 17.58 |
| | + LocAt | $34.94_{+3.56}$ | $49.24_{+0.41}$ | $36.37_{+2.02}$ | $82.54_{+0.38}$ | 86 | 17.64 |
| | Jumbo | 32.20 | 47.31 | 34.65 | 84.42 | 260 | 19.74 |
| | + LocAt | $35.69_{+3.49}$ | $49.20_{+1.89}$ | $35.84_{+1.19}$ | $84.43_{+0.01}$ | 260 | 19.81 |

Table 2: **Classification top-1 accuracy** of ViT and LocAtViT for different backbone sizes on mini-ImageNet and CIFAR-100, showcasing LocAt's effectiveness on small-scale datasets.

| Size | mini-ImageNet | | CIFAR-100 | |
|---|---|---|---|---|
| | ViT | LocAtViT | ViT | LocAtViT |
| Tiny | 74.94 | $78.47_{+3.53}$ | 73.84 | $80.43_{+6.59}$ |
| Small | 78.98 | $84.30_{+5.32}$ | 76.33 | $81.13_{+4.80}$ |
| Base | 79.91 | $84.86_{+4.95}$ | 76.90 | $82.20_{+5.30}$ |

Table 3: **Self-supervised performance of LocAtViT used in DINO**, demonstrating LocAt's effectiveness in the self-supervised regime.

| Experiment | | ViT-S/16 | LocAtViT-S/16 |
|---|---|---|---|
| Linear classification | | 65.52 | $67.65_{+2.13}$ |
| Nearest neighbor | 10-NN | 61.69 | $63.96_{+2.27}$ |
| | 20-NN | 61.53 | $63.74_{+2.21}$ |
| | 100-NN | 59.30 | $61.19_{+1.89}$ |
| | 200-NN | 57.90 | $59.78_{+1.88}$ |

**Foundation models.** In the previous sections, we described our interest in improving ViT's segmentation capabilities without changing their training scheme. Our experiments support that our minor modifications lead to better dense prediction performance, while performing on par or superior to the vanilla models in classification. One reason for our interest in the mentioned problem is that ViTs have been widely used across computer vision foundation models and are the go-to choice for many of the recent methods (Radford et al., 2021; Kirillov et al., 2023; Caron et al., 2021; Oquab et al., 2024). One of the popular models that yields versatile image representations and transfers well to different computer vision tasks is DINO (Caron et al., 2021), which is trained in a self-supervised manner and can serve as a general-purpose backbone.

We train DINO ViT-S/16 and DINO LocAtViT-S/16 on ImageNet-1K for 50 epochs, and evaluate on two tasks used by Caron et al. (2021): learning a linear classifier on top of the frozen backbone and nearest-neighbor classification ($k$-NN) on the features. We train the linear classifier for 50 epochs. Table 3 demonstrates that replacing ViT with LocAtViT in DINO improves its performance on both linear and $k$-NN classification. We report the $k$-NN performance on $k \in \{10, 20, 100, 200\}$

Table 4: **Hummingbird dense nearest-neighbor retrieval** (mIoU %) of models and their counterparts with our LocAt extension for different backbone sizes on PASCAL VOC and ADE20K.

| | Tiny | | | | Base | | | |
|---|---|---|---|---|---|---|---|---|
| | PASCAL | | ADE20K | | PASCAL | | ADE20K | |
| Method | Vanilla | + LocAt | Vanilla | + LocAt | Vanilla | + LocAt | Vanilla | + LocAt |
| ViT | 39.2 | $50.3_{+11.1}$ | 12.0 | $15.2_{+3.2}$ | 55.8 | $58.7_{+2.9}$ | 19.5 | $21.5_{+2.0}$ |
| Swin | 45.2 | $45.3_{+0.1}$ | 16.1 | $16.3_{+0.2}$ | 57.6 | $62.8_{+5.2}$ | 23.3 | $24.6_{+1.3}$ |
| RegViT | 39.4 | $52.3_{+12.9}$ | 12.5 | $15.9_{+3.4}$ | 55.5 | $60.3_{+4.8}$ | 19.4 | $22.8_{+3.4}$ |
| RoPEViT | 50.7 | $54.7_{+4.0}$ | 16.0 | $17.5_{+1.5}$ | 61.0 | $61.4_{+0.4}$ | 22.4 | $23.7_{+1.3}$ |
| Jumbo | 40.0 | $45.5_{+5.5}$ | 13.3 | $14.5_{+1.2}$ | 58.5 | $63.8_{+5.3}$ | 21.6 | $23.7_{+2.1}$ |

as advised by Caron et al. (2021). These findings reveal our objective-agnostic modifications' effectiveness in the self-supervised regime and the potential of our method on backbones that learn general-purpose representations. While interesting, further investigation on larger foundation models is beyond our computational reach and lies outside the scope of this work.

**Hummingbird evaluation.** To further assess whether LocAt improves quality of image features, we evaluate our models using Hummingbird (Balažević et al., 2023), a protocol proposed for evaluating *in-context scene understanding* in a purely frozen-feature regime. We use the implementation by Pariza et al. (2024) and follow its dense nearest-neighbor (NN) retrieval setup. Given a support set of images with semantic segmentation labels, each query image is segmented by retrieving the nearest visual tokens from the support set in the embedding space, without any fine-tuning or decoder training. This protocol therefore measures the intrinsic spatial and contextual quality of representations produced by the backbone, which aligns well with our motivation. Table 4 shows that LocAt consistently improves NN retrieval performance relative to the corresponding vanilla backbones on PASCAL VOC (Everingham et al., 2010) and ADE20K (Zhou et al., 2019) across architectures, suggesting that LocAt enhances spatial representations, even without any task-specific fine-tuning or decoder.

## 5.3 QUALITATIVE ANALYSIS

An interesting implication of our proposed modifications is the refinement of ViT's patch outputs, which makes it more suitable for use cases on dense prediction tasks. Figure 1 offers a visual comparison of attention maps from a vanilla ViT and our LocAtViT, both trained for classification, for an image labeled as *school bus*. From the `[CLS]` token's attention, we observe that ViT's focus is broadly dispersed, whereas LocAtViT shows more concentrated and coherent activation on key features of the bus. Furthermore, we present the attention maps of three patch tokens to other patches. For instance, a patch on the bus side attends to nearly the entire bus in LocAtViT, whereas ViT's map is harder to interpret. A patch covering the child's face generates meaningful attention in both models, but ViT seems to highlight unrelated regions more. Interestingly, for a patch near the top-right corner, LocAtViT not only focuses on some tree patches, but also extends attention to the sky and road, all corresponding to the image background. Despite being trained solely for classification, LocAtViT exhibits an improved ability to detect some scene structures, suggesting that our proposed local interactions can enrich the model's contextual understanding without sacrificing global attention. Further qualitative examples are presented in Appendix C.

## 5.4 ABLATION STUDY

In this section, we provide an ablation study on the architectural choices we made. We also provide an ablation study on the self-attention module's design in the Appendix D.

**Effect of GAug and PRR.** Part ❶ of Table 5 ablates on GAug and PRR defined in Sections 4.1 and 4.2. Results demonstrate that both GAug and PRR indeed enhance the performance of the model in both classification and segmentation, and their combination pushes the performance even further.

Table 5: **Ablation study on model's architecture.** We report segmentation performance (mIoU %) over three benchmarks and classification accuracy (top-1 %) on ImageNet-1K. PE and GAP stand for positional embeddings and global average pooling.

| Method | Tiny | | | | Base | | | |
|---|---|---|---|---|---|---|---|---|
| | ADE | P-Context | C-Stuff | ImageNet | ADE | P-Context | C-Stuff | ImageNet |
| ViT | 17.30 | 33.71 | 20.29 | 72.39 | 28.40 | 43.10 | 30.43 | 80.99 |
| ❶ ViT + GAug | 18.98 | 34.97 | 21.51 | 73.16 | 30.26 | 44.36 | 32.21 | 82.00 |
| ViT + PRR | 21.60 | 37.93 | 25.85 | 73.71 | 29.89 | 44.03 | 32.16 | 82.19 |
| LocAtViT | 23.47 | 38.57 | 26.15 | 73.94 | 32.64 | 45.35 | 33.62 | 82.31 |
| ❷ ViT - PE | 15.13 | 31.94 | 19.35 | 69.36 | 24.59 | 40.18 | 28.79 | 79.39 |
| LocAtViT - PE | 22.69 | 38.15 | 26.05 | 73.10 | 29.73 | 44.69 | 32.17 | 82.17 |
| ViT | 17.30 | 33.71 | 20.29 | 72.39 | 28.40 | 43.10 | 30.43 | 80.99 |
| ❸ ViT + GAP | 19.65 | 34.94 | 22.86 | 72.50 | 27.99 | 41.97 | 29.88 | 81.84 |
| ViT + PRR | 21.60 | 37.93 | 25.85 | 73.71 | 29.89 | 44.03 | 32.16 | 82.19 |

**Effect of positional embeddings.** Part ❷ of Table 5 evaluates the impact of the default absolute positional embeddings (PE) on our proposed LocAt add-on. For both backbone sizes, LocAtViT without PE not only outperforms ViT without PE, but also surpasses ViT with PE. This indicates that LocAt captures the spatial information embedded into PE and more, with much fewer learnable parameters. It is worth noting that our approach is not an alternative to positional encoding and we did not intend to propose a new PE method. Therefore, these results are included just to demonstrate empirically that LocAt indeed captures the spatial information that the default PE captures, which is the mechanism for capturing locality in vanilla ViT. We have shown in Table 1 that LocAt is applicable alongside other, newer positional encoding approaches, such as RoPE, as well.

**Comparison between PRR and GAP.** As discussed in Section 4.2, PRR addresses patches' gradient flow issues while overcoming GAP's limitations in segmentation. Part ❸ of Table 5 compares how vanilla ViT performs when equipped with PRR versus GAP. PRR shows superior segmentation performance and interestingly, it improves classification accuracy more than GAP. Moreover, although GAP helps ViT in classification, it hurts the segmentation performance in the Base backbone, which is in line with the discussions in Section 4.2 about GAP's problems in segmentation.

## 6  CONCLUSION

**Summary.** We present the *Locality-Attending Vision Transformer*, a modular add-on that enhances vision transformers for dense prediction while preserving image-level capabilities and integrating seamlessly into existing ViTs. The approach introduces a segmentation-in-mind pretraining perspective: *GAug* softly biases attention toward local regions to capture fine-grained spatial details, and *PRR* ensures meaningful gradient flow to patch tokens, strengthening representations for dense prediction. Experiments across multiple ViT baselines show that LocAt delivers consistent segmentation performance gains without compromising classification accuracy. Rather than replacing strong existing architectures, we offer a simple, largely orthogonal upgrade for classification-trained ViTs, particularly relevant given their widespread use in foundation models. We hope these minimal changes will be adopted in future ViT-based models.

**Limitations.** We evaluated our method on multiple classification and segmentation benchmarks, but all of them only contain natural images. Extending the evaluation to other domains (*e.g.*, medical imaging or remote sensing) remains future work. Additionally, while we validated LocAt within a small foundation model, evaluating large foundation models (*e.g.*, CLIP-scale) was beyond our computational budget.

### ACKNOWLEDGMENTS

This work was supported by the Natural Sciences and Engineering Research Council of Canada (NSERC) and the Fonds de recherche du Québec – Nature et technologies (FRQNT) under grant no. 369892. We also thank Calcul Québec and Compute Canada for providing the computing resources used in this work.

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

# LOCALITY-ATTENDING VISION TRANSFORMER APPENDIX

## A  TECHNICAL DETAILS

### A.1  CODE AND COMPUTE RESOURCES

We used the implementation provided by Wightman (2019) for ViT (Dosovitskiy et al., 2021), Swin Transformer (Liu et al., 2021), RegViT (Darcet et al., 2024), and RoPEViT (Heo et al., 2024), and we used the official repository of Jumbo (Fuller et al., 2026). Results for all of these methods are reproduced. Jumbo is a new work and its public repository was incomplete at the time of writing this paper, hence we used the available code and implemented some of the components based on the paper. Moreover, the training regime used by Fuller et al. (2026) is more complex than ours (training for more epochs and at different resolutions with separate teachers). We trained Jumbo with a scheme consistent with our other models, except for leveraging distillation similar to Fuller et al. (2026), and we did not change Jumbo's teachers mid-training. Tiny Jumbo models utilize `deit3-base-patch16-224.fb-in22k-ft-in1k`, and Base Jumbo models use `deit3-large-patch16-224.fb-in22k-ft-in1k` as teachers (Touvron et al., 2022). Our experiments were mostly conducted using NVIDIA RTX A6000 48GB, V100 32GB, and A100 40GB GPUs.

### A.2  LLM USAGE

We used LLMs to generate code for plotting figures, tables, and other LaTeX or code-related tasks. We also used LLMs to improve the writing, polish, or shorten the paragraphs, while double-checking the output.

## B  LOCATVIT COMPARISON WITH RELATED WORK

In Table 1, we included five baseline methods and implemented LocAt for each. Table 6 compares LocAtViT to multiple related works from Section 2: CvT-21 (Wu et al., 2021a), Conformer (Peng et al., 2021), ConViT (d'Ascoli et al., 2021), Twins (Chu et al., 2023; 2021), DaViT (Ding et al., 2022), and GCViT (Hatamizadeh et al., 2023). We utilized the `timm` library as well as publicly available code and checkpoints (Wightman, 2019), and evaluated the models on our segmentation pipeline, using the same segmentation protocol as described in Section 5. Although LocAtViT does not achieve the best classification performance, LocAt helps ViT outperform methods like Twins across all three segmentation benchmarks.

Table 6: **Segmentation and classification performance** of Base backbones from prior work and the proposed LocAtViT.

| Method | Segmentation mIoU (%) | | | Top-1 (%) |
| | ADE | P-Context | C-Stuff | ImageNet |
|---|---|---|---|---|
| CvT-21 | 21.40 | 40.91 | 29.29 | 82.50 |
| Conformer | 22.11 | 40.03 | 26.37 | 83.83 |
| ConViT | 23.08 | 44.82 | 25.20 | 82.30 |
| Twins | 30.47 | 44.55 | 32.27 | 82.71 |
| DaViT | 30.68 | 44.87 | 32.38 | **84.64** |
| GCViT | 30.91 | 44.71 | 32.77 | 84.47 |
| LocAtViT | **32.64** | **45.35** | **33.62** | 82.31 |

## C  ADDITIONAL QUALITATIVE EXPERIMENTS

Figure 4 provides three additional images from the mini-ImageNet dataset, alongside the attention maps of the `[CLS]` token and several patches for ViT and LocAtViT.

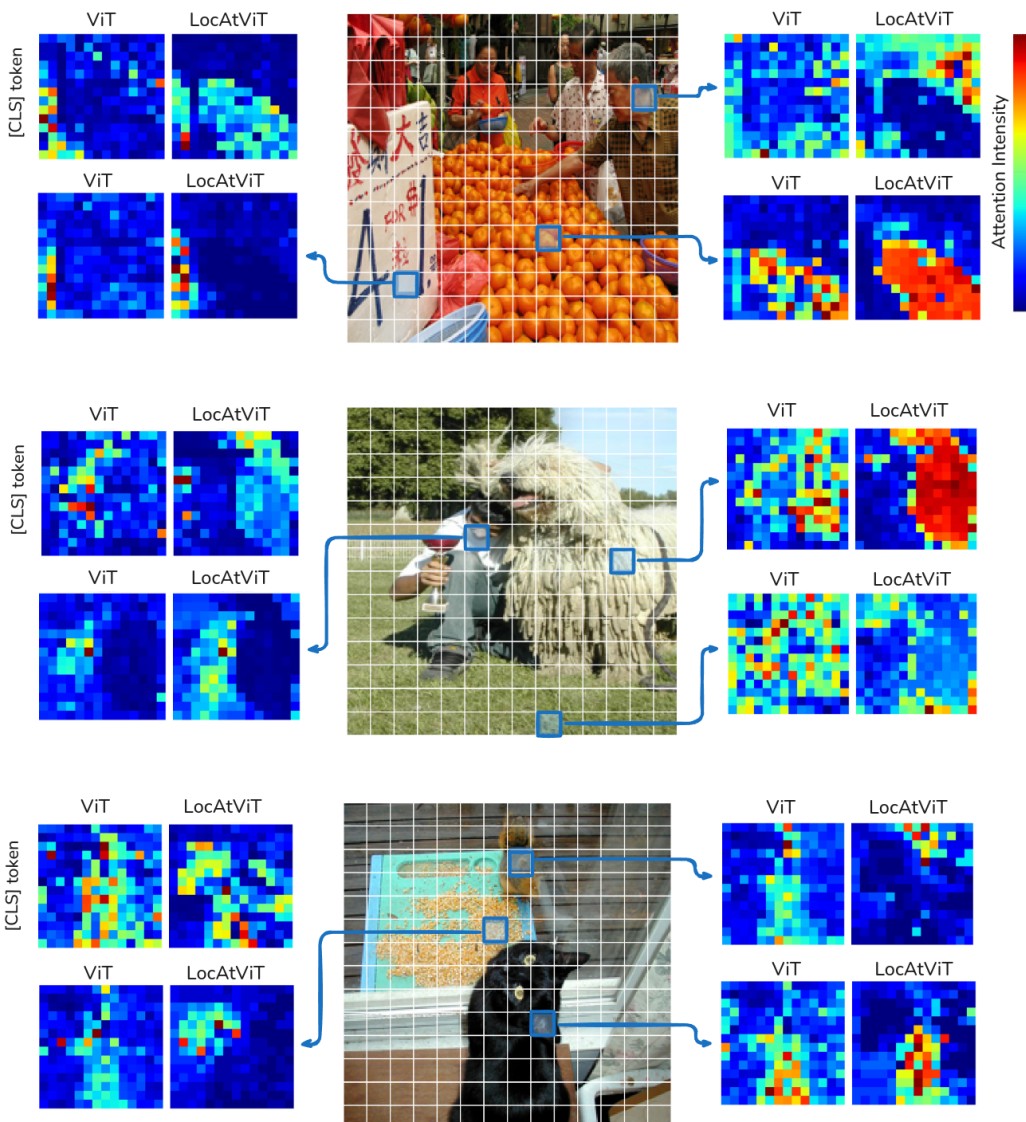

Figure 4: **Qualitative evaluation on the attention maps.** The final attention maps (before the classification head) of ViT and LocAtViT for the `[CLS]` token and three different patches are illustrated for three different images from mini-ImageNet with labels: *orange*, *Komondor*, and *corn*.

## D  ABLATION STUDY ON SELF-ATTENTION

In this section, we perform ablations on the design choices inside the GAug self-attention module.

### D.1  GAUSSIAN BASED ON INPUT

In the original ViT, a query vector intuitively determines the information a patch should be looking for. Since the Gaussian variance controls how far a patch attends to its surroundings, we compute $\Sigma$ based on the spatial query matrix in Eq. (5). Table 7 compares this approach to computing $\Sigma$ based

Table 7: **Ablations on GAug attention components**. $\Delta$#*Params* shows the difference in the number of the parameters of each model compared to LocAtViT (first row). Experiments are conducted on mini-ImageNet, and the classification accuracy (top-1 %) is reported.

|  | Tiny | Base | $\Delta$#Params |
|---|---|---|---|
| LocAtViT (Section 4) | 78.47 | 84.86 | - |
| Gaussian from $\mathbf{x}$ | 79.10 | 85.18 | +18,504, +329,868 |
| Isotropic Gaussian | 78.71 | 84.66 | $-780$ |
| Fixed width $\sigma = 1$ | 75.20 | 82.81 | $-2,340$ |
| $\sigma = 5$ | 76.41 | 82.65 | $-2,340$ |
| $\sigma = 10$ | 75.53 | 82.42 | $-2,340$ |
| No scaling | 76.26 | 83.07 | $-780$ |
| Auto $\boldsymbol{\alpha}$ | 78.48 | 84.54 | $-780$ |

on $\mathbf{x}$, the self-attention input. While the latter improves performance, it significantly increases the number of parameters.

## D.2 VARIANCE MATRIX

To comply with a more general setting, we assigned separate variances for each image axis. An alternative is to use a single variance per patch, forming an isotropic Gaussian kernel. This simplifies Eq. (8) to:

$$\mathbf{G}_{pt} = \exp\Big(-\frac{\sum_{m=1}^{2} \mathbf{D}_{ptm}}{2\sigma_p^2}\Big). \tag{12}$$

The result of this modification is referred to as *Isotropic Gaussian* in Table 7. This table also compares this approach with another experiment where the Gaussian kernel width is fixed to different constant values, instead of being patch-specific and query-based. These results indicate that an isotropic Gaussian kernel performs comparably, but a fixed kernel width substantially diminishes performance, demonstrating the importance of our dynamic input-dependent kernel width.

## D.3 NO SUPPLEMENT MATRIX SCALING

In Section 4.1, we introduced a learnable scaling vector $\boldsymbol{\alpha}$ to match the scale of the supplement matrix $\mathbf{S}$ to that of the attention logits. To isolate its effect, Table 7 reports a variant (*No $\alpha$*) in which the supplement matrix in Eq. (10) is not scaled, *i.e.*, we set $\boldsymbol{\alpha} = \mathbf{1}$. This no-scaling configuration corresponds to a harder use of the locality term and consistently reduces accuracy, confirming that unscaled addition of $G$ is suboptimal and that the learnable scaling is important for balancing global attention with the Gaussian prior.

## D.4 AUTOMATIC SCALING OF THE SUPPLEMENT MATRIX

We motivated the need for scaling the supplement matrix before adding it to the attention logits in Section 4.1. We now propose a parameter-free, input-dependent scheme, *Auto $\boldsymbol{\alpha}$*, that automatically matches the scale of $\mathbf{S}$ to that of the original attention logits. Concretely, define the row-wise $\ell_2$-norm vectors:

$$\mathbf{r} = \big[\|\mathbf{q}_0\|_2, \ldots, \|\mathbf{q}_{hw}\|_2\big]^\top, \tag{13}$$

$$\mathbf{u} = \big[\|\mathbf{k}_0\|_2, \ldots, \|\mathbf{k}_{hw}\|_2\big]^\top. \tag{14}$$

Then the standard attention logits satisfy:

$$\frac{\mathbf{q}\mathbf{k}^\top}{\sqrt{d}} = \Big(\frac{\mathbf{r}\mathbf{u}^\top}{\sqrt{d}}\Big) \circ \cos\big(\mathbf{q}, \mathbf{k}\big), \tag{15}$$

where $\circ$ denotes the Hadamard product, and $\cos(\mathbf{q}, \mathbf{k}) \in \mathbb{R}^{(1+hw) \times (1+hw)}$ has entries $\cos(\mathbf{q}_i, \mathbf{k}_j)$. Hence, if we set:

$$\boldsymbol{\alpha} = \frac{\mathbf{r}\mathbf{u}^\top}{\sqrt{d}} \in \mathbb{R}^{(1+hw) \times (1+hw)}, \tag{16}$$

then the modified logits in Eq. (4) can be rewritten as:

$$\frac{\mathbf{q}\mathbf{k}^\top}{\sqrt{d}} + \mathbf{S} = \boldsymbol{\alpha} \circ \big(\cos(\mathbf{q}, \mathbf{k}) + \mathbf{G}\big), \tag{17}$$

where both terms inside the parentheses are bounded (in $[-1, 1]$ and $[0, 1]$, respectively), ensuring that $\mathbf{S}$ scales comparably to the original logits.

However, using $\boldsymbol{\alpha} \circ \mathbf{G}$ would independently scale each entry of $\mathbf{G}$, destroying the Gaussian kernel structure (each row of $\mathbf{G}$ is a kernel centered at one patch). To preserve each kernel's shape, we average $\boldsymbol{\alpha}$ across columns:

$$\bar{\alpha}_i = \frac{1}{hw} \sum_{j=1}^{hw} \boldsymbol{\alpha}_{ij}, \quad \bar{\boldsymbol{\alpha}} = [0, \bar{\alpha}_1, \ldots, \bar{\alpha}_{hw}]^\top \in \mathbb{R}^{1+hw}, \tag{18}$$

and then form:

$$\mathbf{S} = \mathrm{diag}(\bar{\boldsymbol{\alpha}}) \, \mathbf{G}, \tag{19}$$

similar to Eq. (10). This row-wise scaling applies a single factor to each Gaussian kernel, preserving its shape while matching its magnitude to the attention logits. Unlike the main text, $\mathbf{G}$, $\boldsymbol{\alpha}$, and $\bar{\alpha}$ assume entries corresponding to [CLS] in this section, which should be manually set to zero since [CLS] does not correspond to a spatial location.

Auto $\boldsymbol{\alpha}$ performs close to learnable $\boldsymbol{\alpha}$ in the original LocAtViT, with slightly fewer parameters. We nevertheless keep the learnable $\boldsymbol{\alpha}$ in our main model for simplicity of formulation and to give the network maximal flexibility in attenuating or amplifying locality where beneficial.

## E  ABLATION STUDY ON ALTERNATIVE DISTANCE-BASED KERNELS

In the main text, we model locality with a Gaussian kernel added to the attention logits (Section 4.1). The choice of a Gaussian is motivated by the desire for a smooth, distance-based attenuation function with a scale parameter that controls the effective receptive field, and that can be predicted from each query token. Nevertheless, other monotone distance-based kernels are also reasonable, and we compare against two other kernels in what follows.

Let $r_{pt} = \|P_p - P_t\|_2$ denote the Euclidean distance between patches $p$ and $t$ in the spatial grid. We construct two alternative kernels by predicting scale parameters $\gamma$ and $\lambda$ from the queries:

$$L_{pt} = \exp\left(-\gamma_p \, r_{pt}\right), \tag{20}$$

denoting the Laplace kernel, and the inverse-distance kernel:

$$I_{pt} = \frac{1}{1 + r_{pt}/\lambda_p}. \tag{21}$$

In both cases, the resulting kernel matrix replaces $\mathbf{G}$ in Eq. (10), and the rest of the GAug formulation (including the scaling with $\boldsymbol{\alpha}$) is kept unchanged.

Table 8 compares performance of different choices of the kernel. All three locality-augmented variants improve over the baseline ViT, confirming that introducing a smooth distance-based prior is beneficial. Among them, the Gaussian kernel delivers the strongest segmentation gains on all three benchmarks, while remaining competitive in ImageNet-1K accuracy compared to the Laplace and inverse-distance kernels.

## F  LOCAL FEATURE ANALYSIS ACROSS LAYERS

In the main text, we argue that the global attention mechanism of vanilla ViT tends to obscure fine-grained local information that is important for dense prediction. Here, we provide a quantitative analysis of how local patch features evolve across layers in a standard ViT and in our LocAtViT. We focus on Base models of ViT and LocAtViT trained on ImageNet-1K and evaluate features on the ImageNet-1K validation set.

Table 8: **Effect of different distance-based kernels.** Segmentation performance (mIoU %) over three benchmarks and classification accuracy (top-1 %) on ImageNet-1K are reported.

| Kernel | Tiny | | | | Base | | | |
| --- | --- | --- | --- | --- | --- | --- | --- | --- |
| | ADE | P-Context | C-Stuff | ImageNet | ADE | P-Context | C-Stuff | ImageNet |
| No (ViT) | 17.30 | 33.71 | 20.29 | 72.39 | 28.40 | 43.10 | 30.43 | 80.99 |
| Gaussian | 23.47 | 38.57 | 26.15 | 73.94 | 32.64 | 45.35 | 33.62 | 82.31 |
| Inv-dist | 22.18 | 38.16 | 25.25 | 74.00 | 28.42 | 43.48 | 30.82 | 81.94 |
| Laplace | 21.67 | 37.80 | 25.56 | 74.01 | 29.74 | 44.10 | 31.95 | 82.24 |

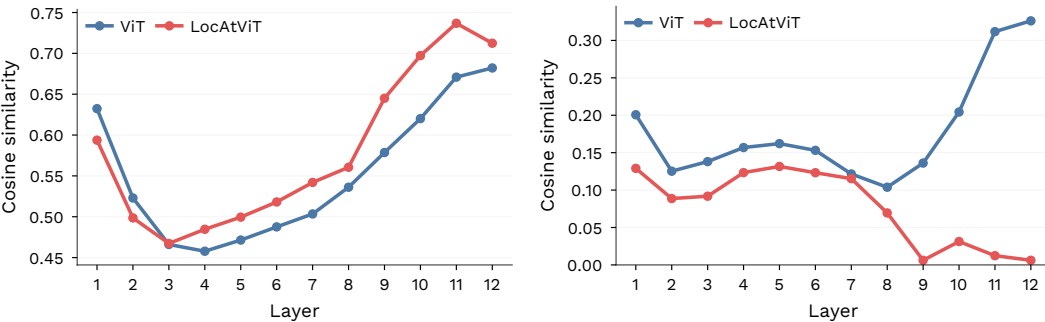

(a) Average cosine similarity to 8 spatial neighbors.   (b) Average cosine similarity to the `[CLS]` token.

Figure 5: **Degradation of local features in vanilla ViT.** Features in ViT collapse to the global information in the last layers while in LocAtViT, patch features encode local information.

**Locality score.**   For each layer $l$ and each spatial patch token, we compute a locality score defined as the cosine similarity between that patch and its 8 immediate neighbors in the surrounding $3 \times 3$ window. We then average this score over all spatial locations and all validation images. Intuitively, a higher locality score indicates that nearby patches share more similar representations, which is desirable as long as representations do not collapse globally. Figure 5a reports this locality score per layer. After the third layer, LocAtViT consistently achieves a higher locality score than vanilla ViT, indicating that its patch features remain more coherent with their spatial neighbors as depth increases.

**Patch-`[CLS]` similarity.**   High neighbor similarity alone does not guarantee that meaningful local structure is preserved: if all patch tokens collapse to the same global representation, their mutual similarity (including to neighbors) will also be high. To distinguish this degenerate case from genuine locality, we additionally measure, for each layer $l$, the cosine similarity between every patch token and the `[CLS]` token, again averaged over all patches and validation images. Figure 5b shows that in vanilla ViT this patch-`[CLS]` similarity steadily increases with depth and peaks in the final layers, revealing a progressive pull of patch features toward a shared global representation dominated by the `[CLS]` token. In contrast, LocAtViT maintains substantially lower patch-`[CLS]` similarity across layers, while still achieving a higher locality score.

**Discussion.**   Taken together, these two measurements show that, in vanilla ViT, patch tokens gradually lose distinct local information and become dominated by global `[CLS]`-like content as depth grows. LocAtViT, on the other hand, preserves strong locality in patch features without collapsing them onto the `[CLS]` token. This behavior aligns with our design goal: to enhance the preservation of local structure while retaining the benefits of global attention, thereby producing representations that are better suited for dense prediction.

# G   STABILITY OF LEARNED STANDARD DEVIATIONS

The per-patch Gaussian variances are predicted from the queries through a bounded nonlinearity in Eq. (5), ensuring numerical stability; however, in principle these values could collapse to the

lower or upper end of the admissible range. Figure 6 analyzes the mean and percentile ranges of the learned standard deviations across layers for a LocAtViT Base model trained on ImageNet-1K. We find that the predicted variances remain well inside the allowed interval and do not cluster near the bounds. These observations indicate that GAug learns meaningful locality scales rather than degenerately switching the Gaussian bias "off" (very small variance) or "fully on" (maximal variance) everywhere.

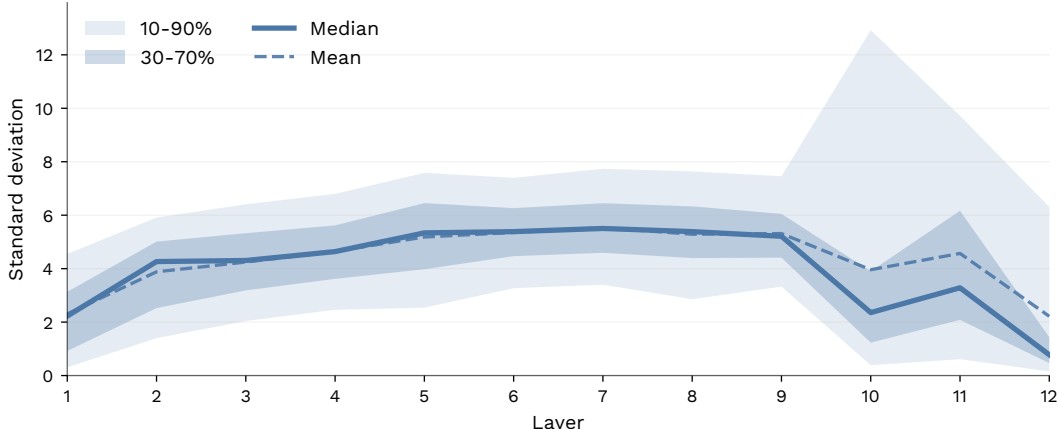

Figure 6: **Layer-wise statistics of the learned Gaussian standard deviation in LocAtViT.** For each layer, we summarize the distribution of learned standard deviation values using percentile ribbons (10–90% and 30–70%) and overlay the median (solid) and mean (dashed).

## H    LIMITATIONS OF THE GAUSSIAN BIAS

Our design goal for the Gaussian augmentation is to gently bias attention toward local structure, rather than to strictly enforce locality. Empirically, across the backbones and tasks reported in the main text, we observe performance gains when adding GAug and PRR. However, the magnitude of the gains depends on the underlying attention topology. The largest improvements appear on backbones with unrestricted patch-patch attention (*e.g.*, ViT, RegViT, RoPEViT, and Jumbo), whereas the gains on a windowed-attention backbone such as Swin are noticeably smaller. This suggests that GAug is most effective when attention is globally connected and locality is not already hard-coded by the architecture.

To further probe this limitation, we also applied our approach to GCViT (Hatamizadeh et al., 2023), a stronger windowed-attention model with attention confined to small grids. In this setting we did not observe improvements in the performance. We attribute this negative result to the fact that when attention is restricted to narrow windows, the additional Gaussian bias has little room to meaningfully reshape the locality pattern. In contrast, even for powerful unrestricted-attention models such as Jumbo, there remains enough flexibility for GAug and PRR to provide noticeable benefits.

