# OpenReview forum: "Locality-Attending Vision Transformer"
_ICLR.cc/2026/Conference — ICLR 2026 Poster_

### Official Review · Reviewer_iq82 · 2025-10-26

**Soundness:** 3
**Presentation:** 3
**Contribution:** 3
**Rating:** 6
**Confidence:** 4

**Summary:**

The paper proposes LocAtViT, a ViT backbone augmentation aimed at making features more locality-aware (useful for dense prediction) while preserving image-level classification strength. Two components are introduced: (i) a Gaussian-biased self-attention that softly favors nearby patches; and (ii) a patch-aware classifier refinement that aggregates patch tokens (rather than relying solely on a [CLS] token or uniform pooling). Experiments (classification on ImageNet-1k; frozen-backbone segmentation on ADE20K, PASCAL-Context, COCO-Stuff; plus a DINO self-supervised setup) show consistent segmentation gains with parity or small gains in classification.

Relative to prior locality work—e.g., ConViT (soft convolutional inductive bias via GPSA) and LocalViT (injecting locality in the FFN)—the paper’s novelty is to impose a learned Gaussian locality bias directly on attention and to pair it with a patch-involving aggregation for classification, both kept lightweight. There is also prior art that explicitly studies Gaussian attention bias in ViTs; LocAtViT’s contribution is its particular way of learning and deploying such a bias for general pretraining and dense transfer.

Summary of the review: The paper proposes a simple, low-cost locality bias for ViTs that improves spatial sensitivity while preserving global context, showing clear gains in frozen-backbone segmentation and self-supervised settings. Its design is elegant, lightweight, and broadly applicable, offering practical value for making ViTs more “segmentation-ready.” However, the evaluation scope is narrow—missing full fine-tuning, detection, and ablations against class-attention or pooling baselines—and more transparency on Gaussian stability would strengthen the case. Overall, this is a solid, well-motivated contribution with practical merit but limited empirical breadth, justifying a rating of 6 for clarity, utility, and moderate originality.

**Strengths:**

1) Simple, plug-in design with low overhead: Aligns with trends showing locality helps ViTs; uses a soft bias so global context remains available.
2) Clear target and protocol: Frozen-backbone segmentation fairly isolates representational gains; positive results in self-supervised DINO suggest generality beyond supervised pretraining.
3) Potential impact: Cost seems negligible and code is clean, thus many ViT backbones could adopt the tweak during pretraining to become more “segmentation-ready”.

**Weaknesses:**

1) Full fine-tuning and detection: Frozen-backbone segmentation is informative but not standard practice; include end-to-end segmentation fine-tuning and at least one object detection benchmark (e.g., COCO with a simple detector) to test if gains persist under typical training.

2) Ablation transparency: Report stability and learned variance scales of the Gaussian (do they collapse or saturate?), and whether the [CLS] treatment or bias masking affects results.

3) Aggregation vs. class-attention / pooling variants. The patch-aware classifier refinement should be positioned and compared to CaiT [1] (class-attention layers) and established token-pooling/labeling approaches; at least show the difference and the relation, or even at least small-scale ablations would help isolate benefits over these alternatives.

[1] Touvron, H., Cord, M., Sablayrolles, A., Synnaeve, G., & Jégou, H. (2021). Going deeper with Image Transformers. arXiv [Cs.CV]. http://arxiv.org/abs/2103.17239

**Questions:**

1) Aggregation baselines: How does your classifier-side refinement compare to CaiT’s class-attention and token pooling / token labeling? Can you add a small table isolating this component?

2) Generalization under full fine-tuning. Do the segmentation gains hold under end-to-end fine-tuning? If yes, please report one such setting (e.g, ADE20K with a standard decoder).

3) Detection transfer: Have you tried COCO object detection (e.g., with a simple Detr head) to check whether locality-biased features help localization beyond conventional semantic tasks?

4) Scaling + overhead: What is the measured training wall-clock and memory overhead for 224x224 → 512x512 inputs?

5) In-Context Visual Understanding: Does the model gain any benefits on the recent proposed Hummingbird evaluation [2] ( implemented openly by [3])?

6) Failure modes: Did you observe maybe whether there are any cases where the Gaussian bias hurt?

[1] Touvron, H., Cord, M., Sablayrolles, A., Synnaeve, G., & Jégou, H. (2021). Going deeper with Image Transformers. arXiv [Cs.CV]. http://arxiv.org/abs/2103.17239
[2] Balažević, I., Steiner, D., Parthasarathy, N., Arandjelović, R., & Hénaff, O. J. (2023). Towards In-context Scene Understanding. arXiv [Cs.CV]. http://arxiv.org/abs/2306.01667
[3] https://github.com/vpariza/open-hummingbird-eval

---

> ### Author Response · Authors · 2025-11-21
> **Response to reviewer iq82 (Part 1/4)**
>
> We thank the reviewer for the careful summary and for emphasizing the strengths of our paper. We are pleased that the **elegance, lightweight nature, and broad applicability** of LocAtViT’s design were appreciated, along with its **simplicity**, and its demonstrated **generality** beyond supervised pretraining. The constructive feedback provided by the reviewer has given us the material to substantially improve our manuscript, and we greatly appreciate that.
>
> ## Q1 & W3 (Aggregation baselines)
>
> ### **CaiT**
>
> CaiT has two main methodological components: *(i)* LayerScale, which stabilizes optimization and enables training very deep ViTs, and *(ii)* class-attention layers, which update only the class token while keeping patch tokens fixed in the last blocks in order to better process the class embedding. The first component is orthogonal to our work and can be directly combined with LocAtViT. The part that is most closely related to our PRR module is the class-attention mechanism, and we focus our comparison there.
>
> **LayerScale helps LocAtViT.**
> Before the main comparison, we would like to empirically support that our approach can also benefit from LayerScale and deep backbones. So, we train on ImageNet-1K a CaiT-XXS-36 model (17M parameters) and a LocAtViT model of the same size that uses LayerScale. CaiT reaches 77.5% top-1 accuracy, whereas the corresponding LocAtViT model achieves 80.2%, showing that LocAtViT effectively leverages LayerScale and performs strongly in the deep setting.
>
> **PRR vs class-attention.**
> We further compare ViT+PRR tiny and base backbones (6M and 86M parameters) against CaiT backbones of similar size in both classification and segmentation:
>
> | Size | Method| ADE  | PC   | Stuff| Acc  |
> |---|---|---|---|---|---|
> | Tiny | PRR  | 21.6 | 37.9 | 25.9 | 73.7 |
> |      | CaiT | 16.9 | 30.2 | 18.7 | 69.6 |
> | Base | PRR  | 29.9 | 44.0 | 32.2 | 82.2 |
> |      | CaiT | 27.8 | 41.9 | 30.1 | 79.1 |
>
> For similar parameter budgets, PRR consistently outperforms CaiT classification and segmentation. We attribute this to a fundamental difference in design and goal:
>
> - CaiT introduces several **parameterized** class-attention blocks inside the backbone, where only the class token attends to patch tokens and patch–patch interactions are removed in these layers. These blocks are tailored to stabilizing very deep ViTs and improving classification, and they deliberately **do not update patch tokens** in the final layers. This design prioritizes the class embedding, but it means the capacity of the last blocks is not used to refine patch features, which is suboptimal for dense prediction.
> - In contrast, PRR is a single **parameter-free** refinement layer attached on top of a standard ViT. It applies full self-attention to all tokens (including patch–patch interactions), so **both the class token and the patch tokens** are updated. This preserves and sharpens local structure while keeping the backbone **architecture faithful to the original ViT design** that is widely adopted in the community. PRR is explicitly motivated by downstream segmentation, where high-quality patch representations are crucial.
> - While CaiT’s design is particularly well suited to very deep models, our experiments suggest that in the parameter and depth regime we study this structural bias is not optimal. In CaiT, the last class-attention blocks exclusively update the [CLS] token while keeping patch tokens fixed, so part of the capacity of the final layers is devoted solely to a relatively shallow refinement of [CLS]. In contrast, ViT+PRR keeps patch–patch and patch–[CLS] interactions active in all backbone layers, utilizing the capacity, and then applies an additional parameter-free refinement over all tokens. We attribute the difference in classification performance of the models under the same training recipe and comparable parameter counts, to the more symmetric use of capacity in our method.
>
>
> ### **Token labeling**
>
> Token labeling modifies the training objective by providing patch-level pseudo-labels from an external teacher, and requires pretraining the backbone with dense supervision. This requires additional annotations or a teacher model, changes the loss, and alters the pretraining regime. In contrast, LocAtViT focuses on lightweight classifier-side refinement that is compatible with popular supervised ViT backbones, does not require additional annotations or teachers, and does not change the training objective and regime. We will clarify this distinction in the related work section. Note that, in principle, token labeling could be combined with our modules, but a full exploration of that combined pretraining regime is beyond the scope of this work.
>
>
> ### **Token pooling**
>
> *Will be discussed in the next comment.*

---

> ### Author Response · Authors · 2025-11-21
> **Response to reviewer iq82 (Part 2/4)**
>
> ## Q1 & W3 (Aggregation baselines) [continue]
> ### **Token pooling**
>
> PRR is intentionally designed for “segmentation-in-mind” pretraining, i.e., good patch features, not just a strong pooled vector. Instead of taking the maximum features as in Max pooling, globally averaging patch tokens as in GAP, or aggregating them via a learnable pooling head such as multihead attention pooling (MAP), PRR applies a single parameter-free self-attention over the final tokens. This operation simultaneously (i) aggregates information in a non-uniform, content-adaptive way and (ii) refines all patch tokens, which is crucial for downstream dense prediction. To compare our method with related token pooling methods, we trained the tiny (6M) variant of Max pooling, AvgMax (a combination of average pooling and max pooling), and multihead attention pooling  (MAP).
>
> | Pooling | ADE  | PC   | Stuff| Acc  |
> |---|---|---|---|---|
> | GAP     | 19.7 | 34.9 | 22.9 | 72.5 |
> | Max     | 19.2 | 34.7 | 23.3 | 71.9 |
> | AvgMax  | 20.1 | 35.6 | 24.2 | 72.3 |
> | MAP     | 20.2 | 36.3 | 23.1 | 73.0 |
> | PRR     | 21.6 | 37.9 | 25.9 | 73.7 |
>
> PRR consistently achieves the best performance across all three segmentation benchmarks and also yields the highest classification accuracy. This demonstrates that PRR provides more informative aggregation while simultaneously improving patch representations, which is precisely what is needed for effective transfer to segmentation tasks.
>
>
> ## Q2 & W1 (Full fine-tuning)
> In our segmentation experiments we deliberately adopted a light-weight decoder with a frozen backbone, and we appreciate that the reviewer believes that this pipeline fairly isolates representational gains. This choice aligns with our goal of isolating and measuring the quality of patch representations in a “black-box” / low-tuning regime (e.g., similar in spirit to training-free or zero-shot dense prediction from foundation ViTs such as CLIP) rather than maximizing absolute mIoU with a heavy head. We believe that a stronger head, full fine-tuning, or much longer training could mask representation differences, which would undermine the clarity of the effect of LocAt.
> That said, we agree that end-to-end fine-tuning with a standard decoder is the typical segmentation setup. To address this, we follow reviewer's suggestion and additionally evaluate full fine-tuning on ADE20K on ViT models. In this setup we use a UperNet head and fine-tune the entire network for 50K iterations, whose results are reported in the following table:
>
> |         | Tiny | Base |
> |---------|------|------|
> | ViT     | 35.7 | 41.2 |
> | + LocAt | 36.9 | 45.2 |
>
> LocAtViT consistently improves mIoU by considerable margins over the backbones even under full fine-tuning. This confirms that the locality bias introduced by GAug and PRR yields more effective representations not only in the frozen-backbone regime, but also in the standard end-to-end segmentation setting.
>
>
> ## Q3 & W1 (Object detection)
> Following the reviewer’s suggestion, we conducted an additional experiment on COCO 2017 object detection and instance segmentation using a Mask R-CNN head with a 1x schedule. We used the Swin tiny models, in two regimes: full end-to-end fine-tuning (FT) and frozen backbone with a trainable head (frozen).
>
> ||mAP^b|mAP^b-50|mAP^b-75|mAP^m|mAP^m-50|mAP^m-75|
> |---|---|---|---|---|---|---|
> | Swin (FT)|42.3|65.0|46.0|38.9|61.9|41.7|
> | + LocAt (FT)|42.8|65.4|46.7|39.3|62.5|42.0|
> | Swin (frozen)|28.9|52.9|28.1|29.3|50.7|30.3|
> | + LocAt (frozen)|29.7|54.2|28.7|30.0|51.6|30.9|
>
> Based on the results, we can confirm that LocAt proves effective in this task as well, underscoring its generalizability.
>
>
> ## W2 (Ablation transparency)
>
> ### **On variance scales of the Gaussian**
>
> In our approach, the per-patch variances are produced via a bounded nonlinearity $f$ applied to $\mathbf{q} \mathbf{W}^\sigma$ (Eq. 5), so in principle their values are stable, but they could still collapse to the lower or upper bound of the allowed range. To assess what actually happens in practice, we track the learned standard deviations of the Gaussian for patches across all ImageNet validation images using the trained LocAtViT-B model. Then, we compute statistics of these standard deviations across layers. Figure 6 (Appendix of the revised version) summarizes the mean and percentile ranges of these standard deviations. We observe that the learned values remain well inside the admissible interval, are not clustered at the lower or upper bound, and form non-trivial patterns across depth (e.g., earlier layers tend to use narrower kernels while deeper layers become broader), rather than collapsing to degenerate minima or saturating at the maximum. This indicates that the Gaussian augmentation learns meaningful locality scales instead of trivially switching off (very small variance) or on (very large variance) everywhere.
>
>
> ### **On [CLS] treatment and “bias masking”**
> *Will be discussed in the next comment.*

---

> ### Author Response · Authors · 2025-11-21
> **Response to reviewer iq82 (Part 3/4)**
>
> ## W2 (Ablation transparency)
>
> ### **On [CLS] treatment and “bias masking”**
>
> Our Gaussian bias is defined purely in the **spatial domain**: for each patch token $i$ with image-plane coordinates $p_i \in \mathbb{R}^2$, we construct a kernel $G$ over other patch locations $j$, where $G_{ij}$ defines the amount added to the logits of the attention from $i$ to $j$. The [CLS] token does not correspond to any spatial location in the image (there is no $p_\text{[CLS]}$), so there is no well-defined notion of a Gaussian “*around*” [CLS]. Consequently, the Gaussian term is only defined for patch–patch pairs, and entries involving [CLS] are identically zero simply because no spatial kernel can be instantiated there.
>
> Implementation-wise, this means that when we embed the spatial Gaussian matrix into the full attention-bias matrix, the row and column corresponding to [CLS] remain zero. This is not an additional architectural trick or masking heuristic, but a direct consequence of restricting the Gaussian augmentation to tokens with valid spatial coordinates. The [CLS] token still attends globally via standard scaled dot-product attention; we simply do not add an ill-defined spatial bias to those logits. This is what we intended to explain in L241 of the original submission. We apologize for the confusion and will clarify this point in the revised manuscript to avoid the impression that the [CLS] handling is an arbitrary masking choice.
>
>
> ## Q4 (Scaling overhead)
> We benchmark memory usage (GB) and wall-clock time (minutes) of training for one epoch on mini-ImageNet dataset for the tiny and base backbone with batch size 16, using one A100 GPU.
>
> | Backbone size | Image side | Wall-clock time | Memory |
> |---|---|---|---|
> | Tiny | 224 | 2.0  | 1.2 |
> |      | 512 | 6.1  | 6.7 |
> | Base | 224 | 3.1  | 4.1 |
> |      | 512 | 21.1 | 25.0|
>
>
> ## Q5 (Hummingbird evaluation)
>
> We thank the reviewer for highlighting the relevance of the Hummingbird evaluation. To address this comment, we integrated our models into the official open-source Hummingbird evaluation pipeline, which measures in-context scene understanding via Dense Nearest-Neighbor (NN) Retrieval using frozen image features. We report mean IoU for semantic segmentation across PASCAL VOC and ADE20K, across every model and backbone size in the main experiments of our paper.
>
> **Pascal VOC:**
> | Size | Backbone| Vanilla | + LocAt |
> |---|---|---|---|
> | Tiny | ViT     | 39.2 | 50.3 |
> |      | Swin    | 45.2 | 45.3 |
> |      | RegViT  | 39.4 | 52.3 |
> |      | RoPEViT | 50.7 | 54.7 |
> |      | Jumbo   | 40.0 | 45.5 |
> | Base | ViT     | 55.8 | 58.7 |
> |      | Swin    | 57.6 | 62.8 |
> |      | RegViT  | 55.5 | 60.3 |
> |      | RoPEViT | 61.0 | 61.4 |
> |      | Jumbo   | 58.5 | 63.8 |
>
>
> **ADE20K**
> | Size | Backbone| Vanilla | + LocAt |
> |---|---|---|---|
> | Tiny | ViT     | 12.0 | 15.2 |
> |      | Swin    | 16.1 | 16.3 |
> |      | RegViT  | 12.5 | 15.9 |
> |      | RoPEViT | 16.0 | 17.5 |
> |      | Jumbo   | 13.3 | 14.5 |
> | Base | ViT     | 19.5 | 21.5 |
> |      | Swin    | 23.3 | 24.6 |
> |      | RegViT  | 19.4 | 22.8 |
> |      | RoPEViT | 22.4 | 23.7 |
> |      | Jumbo   | 21.6 | 23.7 |
>
> Our results show that LocAT consistently improves NN retrieval performance across models, backbone sizes, and datasets. We appreciate that the reviewer has mentioned this relevant evaluation protocol, and will include the results in our paper's Appendix.
>
> ---
> **To be continued...**

---

> ### Author Response · Authors · 2025-11-21
> **Response to reviewer iq82 (Part 4/4)**
>
> ## Q6 (Failure modes)
>
> Our goal with the Gaussian is to bias the model slightly toward local structure and not to hard-enforce locality. In practice, across the experiments we report, we do not observe systematic failure modes where adding the Gaussian bias clearly hurts performance. Architecturally, the risk of such behavior is mitigated by two design choices: *(i)* GAug is added as a residual bias on top of standard dot-product attention, so the model can always fall back to near-global attention if needed; and *(ii)* the learned variances are bounded. A failure we witnessed in earlier internal experiments was related to the fact that we used an unbounded parameterization for $\Sigma$, which could occasionally produce extremely large variances and lead to unstable or suboptimal behavior. In the final version of LocAtViT we replaced this with a bounded parameterization that constrains the variance to a fixed interval. Combined with the residual addition on top of standard attention, this bounded design makes the Gaussian bias a controlled, non-destructive modification.
>
> That said, the Gaussian bias works best when applied to a backbone where patch attention is not restricted, and it has more limited ability to improve windowed-attention methods. As apparent in our results, the gains on unrestricted-attention backbones are substantially larger than on a windowed-attention backbone (Swin). To provide a concrete example where the Gaussian bias could not help, we also applied our approach on top of GCViT and did not obtain improvements. We attribute this unsuccessful attempt to the fact that when attention is already confined to small windows, the additional Gaussian bias is largely redundant and has little room to further shape the locality pattern. Fortunately, we still observe improvements on Swin, but in a stronger windowed-attention model like GCViT our method does not provide additional gains, whereas even powerful unrestricted-attention models like Jumbo continue to benefit noticeably from our add-ons.

---

### Official Review · Reviewer_oh5T · 2025-10-31

**Soundness:** 3
**Presentation:** 3
**Contribution:** 3
**Rating:** 6
**Confidence:** 2

**Summary:**

LocAtViT proposes a simple yet effective way to enhance Vision Transformers with spatial locality by introducing GAug and PRR modules. The method achieves notable segmentation improvements while maintaining classification performance and requires only minimal architectural changes. Its strength lies in practicality and plug-and-play applicability across ViT variants. However, the work lacks a clear problem analysis, a theoretical motivation for the Gaussian kernel, and clarification of the interaction between GAug and PRR, making the approach appear more empirical than principled.

**Strengths:**

1. LocAtViT introduces locality awareness through the GAug and PRR modules with minimal architectural modification.

2. The proposed modules yield substantial improvements on segmentation benchmarks while preserving or even slightly improving ImageNet classification accuracy.

3. The work highlights a valuable perspective that ViT pretraining can be enhanced for dense prediction by refining patch-level representations.

**Weaknesses:**

1. Although the author pointed out that the global attention mechanism of ViT is not conducive to capturing local details, there is a lack of analysis on the degradation of local features in the baseline ViT.

2. The paper directly proposed Gaussian-Augmented attention, but did not explain why a Gaussian kernel was chosen (instead of other forms of local attenuation functions) and its correspondence with human vision or signal attenuation models. The lack of theoretical or empirical support makes this design more like a heuristic attempt.

3. The relationship between GAug and the PRR module was not clarified whether they are coupled or independent. Although LocAtViT is defined as a combination of the two, the paper did not discuss whether there is a complementary or redundant relationship between the two.

**Questions:**

See weakness

---

> ### Author Response · Authors · 2025-11-21
> **Response to reviewer oh5T (Part 1/2)**
>
> We sincerely thank the reviewer for the thoughtful assessment and for highlighting the strengths of our work. We are glad that the **practicality**, **simplicity**, and **efficiency** of LocAtViT is appreciated, as well as its ability to deliver **substantial segmentation improvements**. We also appreciate the recognition of the **valuable perspective** that ViT pretraining can be refined for dense prediction by strengthening patch‑level representations, which we see as a central message of our work.
>
> ## W1 (Degradation of local features)
> We appreciate this comment and agree that an explicit analysis of local feature degradation in vanilla ViT is important. To study this, we analyse ImageNet-1K–pretrained ViT-B and LocAtViT-B and measure how patch features evolve across layers.
> First, we define a feature locality score. For each patch token, we compute the cosine similarity with its 8 immediate neighbors in the surrounding 3×3 window, and average this value across all patches and validation images. Figure 5(a) (Appendix of the revised version) reports this score per layer. We observe that after the third layer, LocAtViT consistently achieves a higher locality score than vanilla ViT, indicating that its patch features remain more similar to their spatial neighbors.
> However, high neighbor similarity alone does not necessarily guarantee that features preserve meaningful local structure: if all patch tokens collapse to the same global representation, their mutual similarity (including to neighbors) will also be high. To distinguish these cases, we additionally measure, for each layer, the cosine similarity between each patch token and the [CLS] token, again averaged over all patches and validation images. As shown in Figure 5(b), this patch–[CLS] similarity in vanilla ViT steadily increases with depth, peaking in the final layers, indicating that patch features are progressively pulled towards a shared, global representation. In contrast, LocAtViT maintains substantially lower patch to [CLS] similarity, while still achieving a higher neighbor locality score.
> Taken together, these results show that in vanilla ViT, patch tokens gradually lose distinct local information and become dominated by global [CLS]-like content, whereas LocAtViT preserves stronger locality in patch features without collapsing them to the global token. This behavior aligns precisely with our design goal for LocAt, i.e., enhancing local feature preservation while retaining the benefits of global attention.
>
> ## W2 (Why a Gaussian kernel)
> We thank the reviewer for raising this point. Our goal is to introduce a smooth, distance-based locality prior that can be seamlessly integrated into standard attention logits, and whose strength and spatial extent can be adapted per token. In this context, a Gaussian kernel is a natural choice for the following reasons.
>
> First, a Gaussian kernel provides a numerically stable, smooth function with monotonic decay of influence with distance, controlled by a single scale parameter ($\sigma$) (or $\sigma_1, \sigma_2$ in the anisotropic case). This gives us a simple and interpretable way to modulate the effective receptive field: small $\sigma$ yields a sharp local focus, while larger $\sigma$ approaches global behavior. Because the kernel is smooth, bounded, and differentiable, adding it to the attention logits avoids discontinuities or hard cutoffs, and ensures stable gradient-based optimization. In our formulation, both the scale $\sigma$ and the overall magnitude $\alpha$ are predicted from the query tokens, making the locality prior **content-adaptive** rather than fixed.
>
> Last, to further clarify this point, we acknowledge that other forms of local attenuation (e.g., Laplace kernel or inverse-distance decays) are also reasonable. Accordingly, we have added a new ablation in here where we **replace the Gaussian kernel with alternative monotone decay functions of distance** while keeping the parameterization and capacity the same, for the tiny backbone size.
> We use the Laplace kernel $\exp(-\gamma_s ||p_s - p_t||)$ and the inverse-distance kernel $1/(1 + ||p_s - p_t||/\lambda_s)$ as substitutes for the Gaussian kernel, with $\gamma_s$ and $\lambda_s$ predicted from the queries.
> Across these variants, we observe that *(i)* all such local priors yield clear improvements over the baseline ViT, and *(ii)* the Gaussian kernel achieves similar classification and better segmentation performance.
>
> We will add the above motivation and the new kernel ablation to the revised manuscript.
>
> | Kernel   | ADE  | PC | Stuff | Acc  |
> |---|---|---|---|---|
> | No (ViT) | 17.3 | 33.7 | 20.3  | 72.4 |
> | Gaussian | 23.5 | 38.6 | 26.2  | 73.9 |
> | Inv-Dist | 22.2 | 38.2 | 25.3  | 74.0 |
> | Laplace  | 21.7 | 37.8 | 25.6  | 74.0 |
>
> ---
>
> **To be continued...**

---

> ### Author Response · Authors · 2025-11-21
> **Response to reviewer oh5T (Part 2/2)**
>
> ## W3 (GAug and PRR relationship)
> Our intention is for these modules to jointly address the same underlying goal (making ViT representations more suitable for dense prediction) at two different points in the pipeline. Although they can be used independently, we view their relationship as complementary rather than redundant, for three reasons:
>
> **Shared goal, different roles.** GAug and PRR are both motivated by the need to improve patch representations for segmentation, but they act at different stages. GAug modifies self-attention inside the backbone to bias information exchange toward local neighborhoods, so that patch tokens can better encode finer spatial details. PRR, in turn, changes how the classifier aggregates tokens, replacing uniform GAP or plain [CLS] usage, with a content-based aggregation that explicitly uses and supervises patch tokens. In other words, GAug makes patch features more local and informative, whereas PRR ensures that these patch features are actually used by the loss and receive gradients.
>
> **Complementarity in practice.** Without PRR, adding GAug to the final block has no effect, because gradients do not effectively propagate to the final layer's GAug weights. I.e., “ViT + GAug in all but the last layer” behaves the same as “ViT + GAug in all layers”. PRR remedies this by distributing gradients to patch outputs, enabling the GAug parameters in the last (and most critical for the output) layer to be learned.
>
> **Empirical evidence of complementarity.** We report separate ablations for ViT (baseline), ViT+GAug, ViT+PRR, and LocAtViT (GAug+PRR) on the same backbones and datasets (Table 4 in the original submission). Both GAug-only and PRR-only variants consistently improve over the baseline on segmentation, which we view as a strength: each module is useful on its own. Crucially, LocAtViT consistently outperforms both single-module variants on all three segmentation benchmarks, demonstrating that their combination yields additional gains. This outcome reflects the complementary nature of the two modules, with GAug enhancing attention locality and PRR ensuring effective gradient routing to patch tokens.

---

### Official Review · Reviewer_bk4S · 2025-10-31

**Soundness:** 3
**Presentation:** 3
**Contribution:** 3
**Rating:** 6
**Confidence:** 3

**Summary:**

This paper proposes the Locality-Attending Vision Transformer, a new attention mechanism aimed at improving the local inductive bias of Vision Transformers (ViTs) without sacrificing global contextual modeling. The key design introduces a modular Locality-Attending (LocAt) add-on, which modulates the attention logits with a learnable Gaussian kernel centered on each query token's location. This work also enhances patch representations for segmentation by introducing minor changes prior to the classification head, preserving the meaningfulness of spatial tokens.

**Strengths:**

+ Conceptually simple but effective modification. In specific, this paper computes a Gaussian kernel based on the query token and incorporates local spatial information into the attention logits, thereby balancing both global and local attention, which is beneficial for segmentation tasks. In addition, to mitigate the limitations of the [CLS] token, this work applies a parameter-free "self-attention"-style refinement to the output of the final layer. Though the solution is simple and straightforward, it is effective.

+ The related work provides a detailed discussion of methods that optimize attention locality.

+ The proposed method yields noticeable improvements across different models and tasks, and the gains observed in the ablation studies are also significant.

**Weaknesses:**

- The paper lacks visualization of the method's effects. For example, case studies showing attention heatmaps after applying Gaussian-Augmented attention and Patch Representation Refinement.

- It is recommended to highlight the motivation for using the Gaussian kernel before the Method section. For example, by listing a table that qualitatively compares convolution-based hybrids, locality mechanisms inside attention, positional encodings, etc., and explicitly points out the advantages of using Gaussian kernel in this work.

- Not all tasks necessarily require explicit local information. Directly adding S to the attention logits is a rather "hard" approach. Would it be possible to introduce a scaling factor $\alpha$ to balance the logits and S, for example, using $\alpha \cdot S$?

- This paper uses a Gaussian term to enhance the locality. As shown in Figure 3, the resulting S matrix appears similar to the pattern of sliding window. The authors are encouraged to clarify how their approach, i.e., Gaussian-Augmented attention differs from sliding window and to explain the performance gain it achieves.

**Questions:**

My concerns are mentioned above.

---

> ### Author Response · Authors · 2025-11-21
> **Response to reviewer bk4S**
>
> We appreciate the reviewer’s summary and recognition of the strengths and contributions of our paper. In particular, we are grateful that the reviewer acknowledges the **simplicity** and **effectiveness** of our approach, which yields **significant performance improvement**, as well as the **comprehensiveness** of the related work. We also thank the reviewer for the critical criticism, which has given us the opportunity to further clarify our motivation and enhance the overall presentation of the work.
>
> ## W1 (Visualization)
> We would like to kindly refer the reviewer to Figures 1 and 4, which present attention maps for multiple patch tokens and the [CLS] token across different images when applying LocAtViT and its ViT counterpart, demonstrating our method's effects.
>
>
> ## W2 (Qualitative comparison)
> Thanks for the interesting suggestion. We now provide a qualitative comparison table that contrasts existing locality mechanisms and motivates our proposed Gaussian-based approach:
>
> | Mechanism family | Changes backbone architecture? | Locality type | Easily applicable on ViT architecture? | Query-adaptive locality? |
> |---|---|---|---|---|
> | Conv-based hybrids | Yes (conv stems / stages) | Strong local bias via fixed kernels | No (requires specific architectural changes and design choices)  | No (kernel is fixed after training) |
> | Local window / block attention | Yes (window partitioning etc.) | Hard locality within windows; restricted cross-window links | No (needs windowed attention structure) | Partial (attention is content-based within a fixed window) |
> | Positional encodings | No | Implicit spatial bias; no explicit distance-based decay | Yes | No (positional bias is not query-dependent) |
> | **Gaussian-Augmented attention (ours)** | No | Soft locality via continuous and smooth decay | Yes (adds on top of standard attention logits) | Yes (Gaussian parameters predicted from each query) |
>
> We have placed this table before the Method section, as recommended, to highlight the motivation for our method. We summarize the properties in the Table and provide detailed explanations in the Appendix, section E.
>
>
> ## W3 (Scaling factor)
> We agree that it is important not to impose an overly "hard" locality bias and to control the relative contribution of the Gaussian prior. In fact, the current method already introduces exactly such a scaling factor $\alpha$ to balance the original attention logits and the supplement matrix. Concretely, as described in Sec. 4.1, we do not use $S = G$ directly. Instead, we predict a row-wise scaling vector $\alpha = \operatorname{softplus}(q W_\alpha)$ from the query tokens and form $S = \operatorname{diag}(\alpha) G$ (Eq. 9-10) before adding $S$ to the logits. Intuitively, $\alpha$ acts as a learned balancing factor between the original attention term and the Gaussian locality prior: when $\alpha$ is small, the behavior approaches vanilla attention (weak locality), and when $\alpha$ is larger, the locality bias becomes stronger. We kindly believe that this addresses the reviewer’s concern about a systematic "hard" addition of $S$.
>
> We also empirically analyze the effect of scaling in Appendix D.3–D.4. In the "No $\alpha$" variant, we set $S=G$ (i.e., $\alpha=1$) and observe a drop in accuracy, confirming that unscaled addition of $G$ is indeed suboptimal, as intuitively suggested by the reviewer. In Appendix D.4, we further propose a parameter-free Auto-$\alpha$ scheme that automatically matches the magnitude of $S$ to that of the original logits and show that it performs close to the learnable $\alpha$ used in LocAtViT.
>
>
> ## W4 (Comparison with sliding window)
> For illustrative purposes, the matrix $S$ shown in Figure 3 is simplified and does not reflect the actual values used during training. In fact, the displayed pattern corresponds to the special case where $\Sigma$ and $\alpha$ are equal to 1 for all patches. Regardless of this simplified example, Gaussian-Augmented attention is fundamentally different from standard sliding-window attention in how it is applied and in the flexibility it provides.
> Sliding-window attention enforces a hard structural constraint: each query is restricted to attend only to keys within a fixed spatial window (its size is a hyperparameter), with all logits outside the window effectively set to $-\infty$. This changes the attention topology and removes long-range connections by design. Besides, the span of attention (window size in this case) is not content-adaptive. In contrast, our Gaussian-Augmented attention keeps the full global attention graph and introduces a soft bias. All entries of $S$ remain finite, so no key is ever masked out, distant tokens simply receive a smaller prior weight rather than being forbidden. Furthermore, both the magnitude $\alpha$ and the effective width of the Gaussian (through $\Sigma$) are predicted from each query, making the locality content-adaptive.

---

### Comment · Area_Chair_2fwy · 2025-11-23
**The authors' rebuttal is available. Please read, comment, and discuss.**

Dear Reviewers,

Thanks for your time and effort in reviewing ICLR2026 submissions. The authors have provided their responses to your review. Please read and raise your further comments, and discuss with the authors.

Best regards,

Your AC

---

> ### Author Response · Authors · 2025-11-28
>
> Dear Reviewers,
>
> Thank you so much for your constructive feedback and the time you dedicated to reviewing our paper. We kindly invite you to review our responses and let us know if any of your concerns remain unaddressed, so that we can further clarify and discuss those points.
>
> Sincerely,
>
> The Authors

---

### Author Response · Authors · 2025-12-03
**Summary of rebuttal contributions**

We thank the Area Chairs and reviewers for their careful evaluation of our work. All three reviewers expressed positive views about our work (*simple, effective, elegant, valuable perspective, practical, broadly applicable, significant performance improvement, etc.*), assigned pre-rebuttal *scores of 6*, and rated the *soundness, presentation, and contribution* of the paper as *good*. Although scores were not updated after the rebuttal (due to the OpenReview incident), we addressed all points raised in the reviews in our rebuttal and revised the manuscript accordingly. New and modified content is highlighted in blue in the revised version.

Below we summarize the main changes, organized by reviewer, and indicate where they appear in the paper.


### Reviewer bk4S

1. (W1) **Visualizations**: The requested visualizations are presented in *Figures 1 and 4*.
2. (W2) **Qualitative comparison table**: The requested table has been added in the suggested location (*Table 1, before the Method section*), with further details in *Appendix E*.
3. (W3) **Scaling factor**: The setting requested by the reviewer is already present in the original submission, in *Eqs. 9-10* and empirically evaluated in *Section D.3*. We added clarifications at the end of *Section 4.1* to better elaborate on this design choice and rephrased *Section D.3* to avoid future confusion.
4. (W4) **Sliding window**:
We clarified the fundamental differences between our approach to sliding window in response to the reviewer and added comparisons in *Table 1*. Since the reviewer identified the matrices in *Figure 3* as a source of the confusion, we now explicitly state in the *figure caption* that the matrix values are simplified for illustration.


### Reviewer oh5T

1. (W1) **Degradation of local features in ViT**: We added a quantitative analysis of this degradation in *Appendix G* and briefly discuss it in *the second paragraph of Introduction*.
2. (W2) **Why a Gaussian kernel**: We included a discussion and empirical validation in *Section 4.1, the modified self-attention paragraph* and *Appendix F*.
3. (W3) **GAug and PRR relationship**: We clarified the complementarity of these components in the rebuttal (now incorporated at the end of *Section 4* in the revised manuscript), along with empirical evidence in *Section 5.4, the effect of GAug and PRR paragraph*.


### Reviewer iq82

1. (Q1 & W3) **Aggregation baselines**: We expanded the related work discussion in *Section 2, improving token representation paragraph*. We also added pooling baseline results to *Appendix H* and CaiT results to *Appendix I*.
2. (Q2 & W1) **Full fine-tuning**: The results are reported in *Appendix J*.
3. (Q3 & W1) **Object detection**: We added this experiment in *Appendix K*.
4. (W2) **Ablation transparency**: For [CLS] bias masking, we clarified in the rebuttal that the [CLS] handling is not an arbitrary masking choice and modified the lines that led to that confusion (*below Eq. 8*). Regarding the analysis of standard deviation scales, we added it to *Appendix L*.
5. (Q4) **Scaling overhead**: The report is added to *Appendix M*.
6. (Q5) **Hummingbird evaluation**: We added to this experiment to the main text (*just before Section 5.3*) as this evaluation protocol aligns well with our goal of assessing the quality of patch representations.
7. (Q6) **Failure modes**: We added a discussion to *Appendix N*.

We are grateful to the reviewers for their constructive feedback, which has helped us strengthen our paper, and we believe that our rebuttal and the revised manuscript address the raised concerns.

---

### Meta-Review · Area_Chair_Z65U · 2026-01-07

**Summary:**

In the initial phase, this paper received positive scores (6,6,6) from three experts in the field. While advantages of the paper (simplicity, good performance, potential impact, etc.) were recognized, reviewers raised several concerns.

Reviewer bk4S: lack of visualization, highlighting the motivation, the significance of Gaussian-Augmented attention.
Reviewer oh5T: lack of analysis on the degradation of local features, motivation of Gaussian kernel, missing details of proposed modules.
Reviewer iq82: lack of full fine-tuning experiments, lack of ablation transparency, lack of comparison with CaiT.

Overall, the concerns of reviewers center on writing, presentation and additional experiments.

**Reviewer Concerns:**

The authors addressed the concerns of Reviewer bk4S by providing visualisations, more experiments and analysis.
The authors addressed the concerns of Reviewer oh5T by giving more analysis, explanation, and details on the proposed modules.
The authors addressed the concerns of Reviewer iq82 by conducing additional experiments (full fine-tuning, new comparison) and providing more ablation results.

In addition, the main paper was modified accordingly.

**Reviewer Scores:**

Since all the concerns were addressed, it is very likely that the reviewers would either raise their scores or keep their positive scores.

The AC agrees with the reviewers in the strengths of the paper, which could have a broad impact on the community. Therefore, the recommendation is to accept the submission.

---

### Decision · Program_Chairs · 2026-01-26

Accept (Poster)